# Engineered bacterial voltage-gated sodium channel platform for cardiac gene therapy

Hung X. Nguyen[1,4], Tianyu Wu [1,4], Daniel Needs[1], Hengtao Zhang[1], Robin M. Perelli[2,3], Sophia DeLuca [3], Rachel Yang [1], Michael Pan[1], Andrew P. Landstrom [2,3], Craig Henriquez[1] & Nenad Bursac [1✉]

Therapies for cardiac arrhythmias could greatly benefit from approaches to enhance electrical excitability and action potential conduction in the heart by stably overexpressing mammalian voltage-gated sodium channels. However, the large size of these channels precludes their incorporation into therapeutic viral vectors. Here, we report a platform utilizing small-size, codon-optimized engineered prokaryotic sodium channels (BacNa$_v$) driven by muscle-specific promoters that significantly enhance excitability and conduction in rat and human cardiomyocytes in vitro and adult cardiac tissues from multiple species in silico. We also show that the expression of BacNa$_v$ significantly reduces occurrence of conduction block and reentrant arrhythmias in fibrotic cardiac cultures. Moreover, functional BacNa$_v$ channels are stably expressed in healthy mouse hearts six weeks following intravenous injection of self-complementary adeno-associated virus (scAAV) without causing any adverse effects on cardiac electrophysiology. The large diversity of prokaryotic sodium channels and experimental-computational platform reported in this study should facilitate the development and evaluation of BacNa$_v$-based gene therapies for cardiac conduction disorders.

[1] Department of Biomedical Engineering, Duke University, Durham, NC, USA. [2] Department of Pediatrics, Division of Cardiology, Duke University School of Medicine, Durham, NC, USA. [3] Department of Cell Biology, Duke University School of Medicine, Durham, NC, USA. [4] These authors contributed equally: Hung X. Nguyen, Tianyu Wu. ✉email: nbursac@duke.edu

D
ue to the critical roles of voltage-gated sodium channels (VGSCs) in action potential initiation and conduction, genetic mutations that decrease sodium current can cause reduced tissue excitability, leading to various neuronal, cardiac, and skeletal muscle disorders[1]. In addition to mutations that directly alter VGSCs, reduced-sodium current density and slow action potential conduction can arise from the altered extracellular environment, cell morphology, or channel regulation that occur in acquired pathological conditions, such as cardiac ischemia, infarction, and failure[2–4]. The ability to augment sodium current amplitude and tissue excitability via exogenous expression of functional VGSCs thus holds significant therapeutic potential for a variety of cardiac diseases. However, gene-based therapies involving VGSCs are largely hampered by the inability to stably express mammalian channels using adeno-associated virus (AAV) as their large (>6kb) genes exceed the AAV packaging limit. In contrast, bacterial sodium channels (BacNa$_v$s)[5–8] are encoded by genes that are only ~0.7–0.9 kb in size, making them suitable for packaging into any type of recombinant viral vector. Despite this advantage, BacNa$_v$ channels have mostly been utilized as models to study the structure, gating mechanisms, and pharmacology of mammalian VGSCs rather than as therapeutic substitutes[9–11].

Previously, we demonstrated that two engineered BacNa$_v$ variants (Na$_v$RosD G217A and Na$_v$SheP D60A) could be combined with the inward-rectifier potassium channel (K$_{ir}$2.1, gene *KCNJ2*) and the connexin-43 gap junction (Cx43, gene *GJA1*) channel to generate electrically excitable and actively conducting somatic cells[12,13] capable of functionally bridging large conduction gaps in excitable tissues[12]. In this current report, we sought to further explore the therapeutic suitability of engineered BacNa$_v$ channels for gene therapy applications by optimizing their membrane expression and investigating their effects on the excitability and conduction in cardiac tissues from different species, in vitro, in silico, and in vivo. Specifically, we demonstrate that optimized viral expression of BacNa$_v$ significantly improves excitability and velocity of action potential conduction in rat and human cardiomyocyte cultures without altering endogenous ion currents and greatly decreases the incidence of reentrant arrhythmias in an in vitro model of fibrotic cardiac tissue. In silico cross-species studies reveal that these improvements in cardiomyocyte excitability and conduction can also be realized in adult ventricular myocytes and tissues with impaired excitability and pathological structure. Finally, stable virally induced expression of an optimized version of Na$_v$SheP D60A variant (h2SheP) and its effects on cardiac electrophysiology are demonstrated in mouse hearts in vivo. Collectively, these results warrant the future development of the BacNa$_v$ gene therapy platform towards cardiac arrhythmia applications.

## Results

### Improving BacNa$_v$ protein translation via codon optimization.
Because the translational machinery in different organisms exhibits bias towards usage of specific codons[14], we tested if different codon optimization schemes could improve the translation efficiency of the previously characterized Na$_v$SheP D60A[7,12] sequence (bSheP) in human cells. Specifically, we compared expression levels of bSheP to the codon-optimized sequences from Genscript (hSheP) and ATUM (h2SheP) by generating bicistronic lentiviral constructs with each sequence linked to GFP via the viral T2A peptide and driven by the CMV promoter. Due to the ribosome-skipping mechanism of viral 2A peptides[15], any change in transcriptional and translational efficiency of the Na$_v$SheP D60A gene as a result of codon optimization was expected to change GFP expression level. Under

similar transduction efficiencies, K$_{ir}$2.1-expressing monoclonal HEK293 line transduced with bSheP-2A-GFP virus yielded lower GFP intensity compared to transductions with either hSheP-2A-GFP or h2SheP-2A-GFP virus (Fig. 1a–c), with ATUM optimized construct showing the highest GFP signal (Fig. 1d). Patch-clamp recordings of sodium current (I$_{Na}$) revealed consistent trends for Na$_v$SheP D60A expression, with hSheP and h2SheP exhibiting 3.3-fold and 5.4-fold higher peak I$_{Na}$ ($-204 \pm 29$ and $-312 \pm 17$ pA/pF), respectively, compared to non-optimized bSheP ($-58 \pm 5$ pA/pF) (Fig. 1e–f). As a result, maximum AP upstroke velocity with bSheP ($66 \pm 10$ V/s) was improved 2.5 and 3.8 times using hSheP ($168 \pm 36$ V/s) and h2SheP ($252 \pm 28$ V/s), respectively (Fig. 1g–h). As inward Na$_v$SheP current contributes to the plateau phase of AP, an increasing trend in AP duration (APD$_{80}$) was observed with higher I$_{Na}$, albeit without any statistically significant difference (Fig. 1i). Resting membrane potential also remained stable across all three groups (Fig. 1j). These results demonstrated significant improvement in the expression level of functional Na$_v$SheP D60A channels via codon optimization, particularly using the ATUM algorithm, and h2SheP was thus selected for all subsequent studies.

### Optimizing cardiomyocyte-specific BacNa$_v$ expression.
Regarding the potential use of BacNa$_v$ in cardiac gene therapy would benefit from robust cardiomyocyte (CM)-specific expression of the channel, we sought to replace the ubiquitous CMV promoter with a strong myocyte-specific promoter. Using cocultures of neonatal rat ventricular myocytes (NRVMs) and fibroblasts, we compared the effects of lentiviral expression of h2SheP-2A-GFP driven by a cardiac troponin T (cTnT) promoter[16] or a hybrid MHCK7[17] promoter. Transduction with each lentivirus in NRVM-fibroblast cocultures yielded strictly cardiomyocyte-specific expression (Supplementary Fig. 1a, b), with cTnT-h2SheP-2A-GFP lentivirus resulting in notably lower GFP intensity (Fig. 2a) and h2SheP mRNA level (Fig. 2d) compared to CMV-h2SheP-2A-GFP (Fig. 2b, d) and MHCK7-h2SheP-2A-GFP (Fig. 2c, d) lentiviruses. Optical mapping of transduced NRVM cocultures showed a consistent trend, with MHCK7-h2SheP-2A-GFP and CMV-h2SheP-2A-GFP yielding 1.2–1.4- and 1.8–2-fold higher conduction velocity (CV) compared to cTnT-h2SheP-2A-GFP and GFP-only group, respectively (Fig. 2e and Supplementary Fig. 1c-f). No statistically significant differences were observed in APD$_{80}$ (Fig. 2f) and maximum capture rate (MCR, Fig. 2g); however, a slight increasing trend in APD$_{80}$ could be observed for the faster-conducting groups (Fig. 2f). The MHCK7 promoter was thus selected for all subsequent studies due to its robust CM-specific expression profile.

### Effects of BacNa$_v$ expression in excitable HEK293 lines.
To achieve consistent outcomes with overexpressing h2SheP channels, it is critical that their presence in excitable cells does not influence the expression or function of endogenous channels. We thus examined the effects of h2SheP expression in our established engineered excitable cell line Ex293[18–20] by generating a stable derivative monoclonal cell line (ExSheP293) co-expressing h2SheP with the pore-forming α-subunit of the voltage-gated cardiac sodium channel (Na$_v$1.5, gene *SCN5A*), K$_{ir}$2.1, and Cx43. The presence of the h2SheP in these cells did not alter the expression of the three-channel genes (SCN5A, KCNJ2, and GJA1) (Fig. 3a) or density of Na$_v$1.5 (Fig. 3b) and K$_{ir}$2.1 (Fig. 3c) currents. Thus, any changes in the electrical and conduction properties due to h2SheP expression were attributable to this channel alone. To more rigorously examine h2SheP-specific effects on cell excitability and conduction, we applied

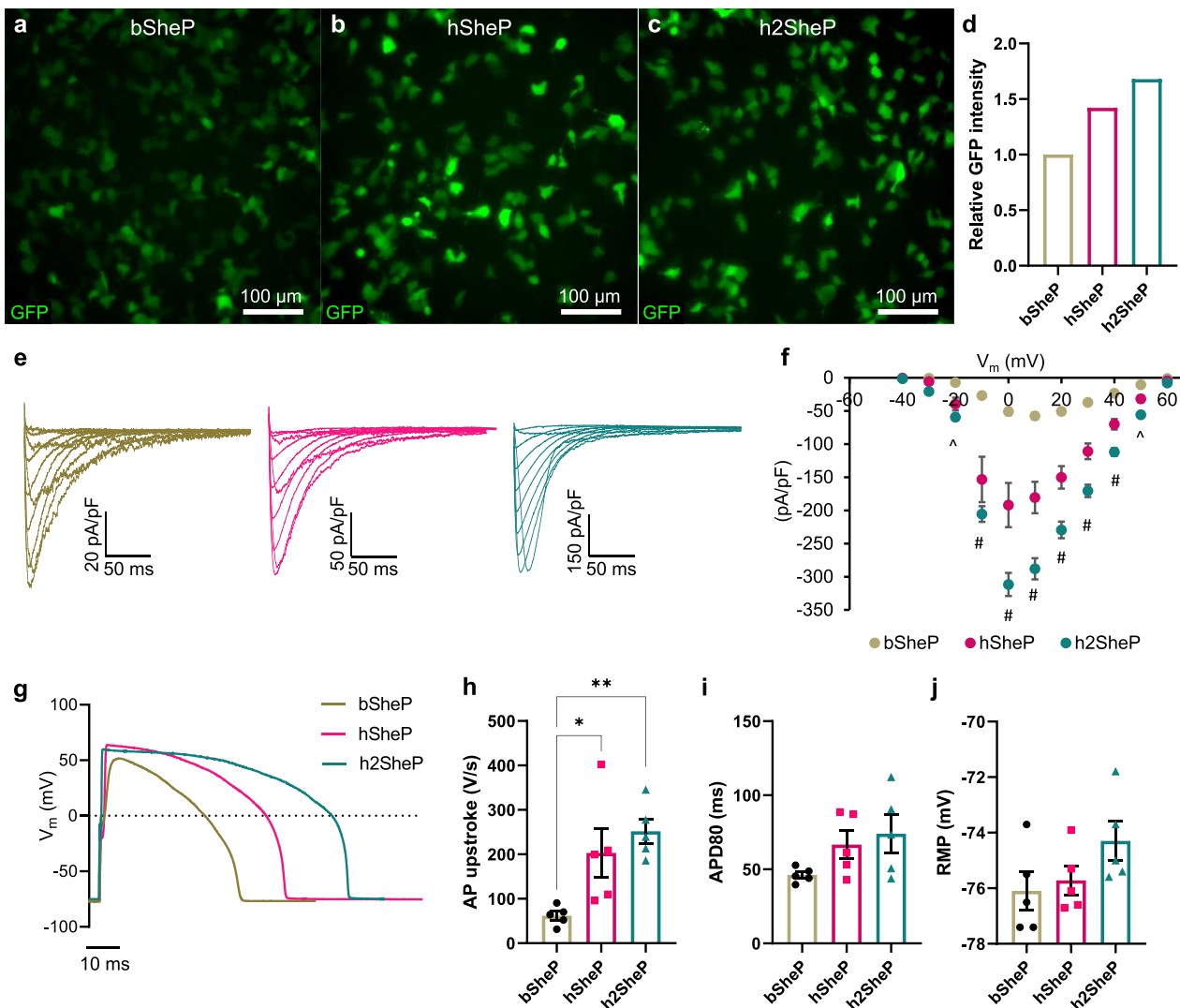

**Fig. 1 Human codon optimization of BacNa$_v$ gene improves expression of functional channels. a–d** Representative images of HEK293 cells transduced with bicistronic lentiviruses in which GFP gene was linked via T2A peptide with non-optimized (bacterial) Na$_v$SheP D60A sequence (bSheP, **a**) or Na$_v$SheP D60A sequences codon-optimized using Genscript (hSheP, **b**) or ATUM (h2SheP, **c**) algorithms and corresponding quantification by flow cytometry (**d**). **e**, **f** Representative current traces (**e**) and corresponding quantifications of peak I$_{Na}$–V curves (**f**) recorded in bSheP, hSheP, or h2SheP-expressing HEK293 cells using whole-cell voltage clamp at 25 °C ($n = 6$). **g–j** Representative action potential (AP) traces (**g**) measured via current clamp in BacNa$_v$-transduced K$_{ir}$2.1-expressing HEK293 cells and corresponding quantifications of maximum upstroke velocity (**h**, AP upstroke; $n = 5$), AP duration at 80% repolarization (**i**, APD80; $n = 5$), and resting membrane potential (**j**, RMP; $n = 5$), all recorded at 37 °C. #$P < 0.05$ among all three groups and ^$P < 0.05$ for h2SheP vs. bSheP in **f**, exact $P$-values for all groups are included in Source Data. *$P = 0.0403$, **$P = 0.0073$ vs. bSheP in **h**. Error bars indicate s.e.m; statistical significance was determined by two-way ANOVA in **f** and one-way ANOVA in **h**, followed by Tukey's post-hoc test to calculate $P$-values. Source data are provided as a Source Data file.

tetrodotoxin (TTX) to selectively block the mammalian Na$_v$1.5 current, while no effects on prokaryotic sodium channels were expected[5]. Consistent with previous studies[18], increasing TTX concentration gradually reduced Na$_v$1.5 current leading to a complete block at 50 µM TTX (Fig. 3d, Supplementary Fig. 2a), while h2SheP current remained stable at all TTX doses (Fig. 3e, Supplementary Fig. 2a). The increasing TTX doses applied to a monolayer of Ex293 cells yielded progressive conduction slowing with a complete block at 10 µM TTX (Fig. 3f), while in the case of ExSheP293 cells, this block was prevented presumably due to the expression of TTX-insensitive h2SheP (Fig. 3g). Simultaneously, APD$_{80}$ was only reduced for the highest TTX concentration (Supplementary Fig. 2c). To further prove that altering CV and APD of Ex293 and ExSheP293 cells by TTX was a consequence of the selective Na$_v$1.5 blockade, we generated a monoclonal line

stably expressing K$_{ir}$2.1, Cx43, and h2SheP (KirCxSheP293). Since in this case the TTX-insensitive h2SheP was the only depolarizing current, both CV (Fig. 3h) and APD$_{80}$ (Supplementary Fig. 2d) of KirCxSheP293 remained stable across all tested TTX concentrations. The higher CV in the KirCxSheP293 vs. the ExSheP293 line at 10 µM TTX was likely a consequence of stronger h2SheP expression.

Considering that the inactivation kinetics of BacNa$_v$ current is relatively slow, we also compared h2SheP channels to Na$_v$1.5 channels treated with anemone toxin ATX II, which delays Na$_v$1.5 inactivation to induce persistently, "late" I$_{Na}$[21,22]. Enhanced late I$_{Na}$ in CMs has been strongly implicated in atrial and ventricular arrhythmogenesis in humans[23,24]. To compare time-courses of h2SheP and Na$_v$1.5 currents under the same AP conditions, we applied a simulated human ventricular AP[25] as the

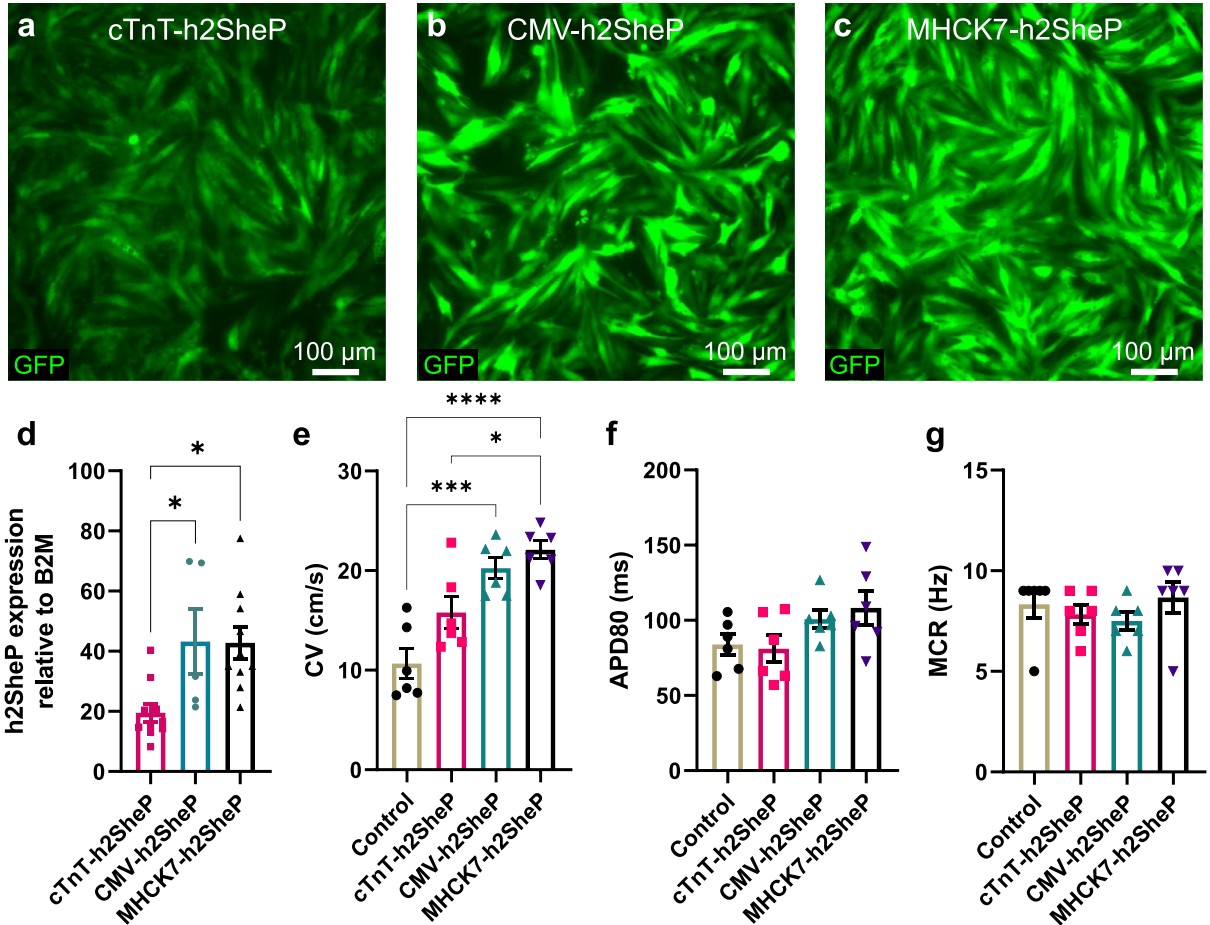

**Fig. 2 Optimization of BacNa_v expression in cardiomyocytes via promoter selection. a–c** Representative images of NRVM monolayers transduced with h2SheP-T2A-GFP lentiviruses driven under cTnT (**a**), CMV (**b**), or MHCK7 (**c**) promoter. **d** Relative mRNA expression of the h2SheP gene normalized to housekeeping gene B2M, quantified in NRVMs transduced with specified lentiviruses ($n = 5$ for cTnT-h2SheP group, $n = 10$ for CMV- and MHCK7-h2SheP groups). *$P = 0.0346$, CMV-h2SheP vs. cTnT-h2SheP; *$P = 0.0103$, MHCK7-h2SheP vs. cTnT-h2SheP. **e–g** Average conduction velocity (**e**, CV), APD80 (**f**), and maximum capture rate (**g**, MCR) values determined during optically mapped AP propagation in NRVM monolayers transduced with a CMV-GFP lentivirus (Control) or specified h2SheP lentiviruses ($n = 6$). *$P = 0.0155$, MHCK7-h2SheP vs. cTnT-h2SheP; ***$P = 0.0003$, CMV-h2SheP vs. Control; ****$P < 0.0001$, MHCK7-h2SheP vs. Control in **e**. Error bars indicate s.e.m; statistical significance was determined by one-way ANOVA, followed by Tukey's post-hoc test to calculate $P$-values. Source data are provided as a Source Data file.

command potential in the voltage-clamp mode (i.e., AP-clamp) to Na_v1.5-expressing Ex293 cells without and with 100 μM ATX II and to h2SheP-expressing KirCxSheP293 cells (Supplementary Fig. 3a–c). Compared to KirCxSheP293 cells where the BacNa_v current turned off in less than 50 ms, in Ex293 cells, the ATX II-induced late Na_v1.5 current persisted and increased in amplitude during late AP repolarization (Supplementary Fig. 3a-e), consistent with the previous studies[22]. In the current-clamp mode, abnormally long APs were only observed in the presence of ATX II-induced late $I_{Na}$[26,27] but not untreated h2SheP channels (Supplementary Fig. 3f-h), suggesting that h2SheP current in human CMs would be unlikely to have pro-arrhythmic effects akin to increased late Na_v1.5 current.

**Effects of BacNa_v expression on cardiomyocyte electro-physiology in vitro.** We next assessed the effects of h2SheP expression in cultured CMs which exhibit significantly more complex electrophysiology than Ex293 cells due to expressing a larger set of membrane ion channels and transporters. Consistent with our results in the Ex293 line, stable lentiviral h2SheP over-expression in NRVMs did not alter mRNA level of the

endogenous Na_v1.5 channel (Fig. 4a) or any other common cardiac ion channels and transporters (Supplementary Fig. 4). Patch-clamp recordings further demonstrated no change in endogenous Na_v1.5 current density due to h2SheP or GFP (control) transduction (Fig. 4b, c). Moreover, h2SheP-transduced NRVMs exhibited robust h2SheP current (Fig. 4d) with the characteristic peak current-voltage relationship (Fig. 4e). This robust h2SheP expression yielded more than 1.5-fold increases in AP upstroke velocity (Fig. 4f, g) and 1.2-fold increases in AP amplitude (APA, Fig. 4h) compared to control groups, with no changes in APD_80 (Fig. 4i) or resting membrane potential (RMP, Fig. 4j). To further demonstrate the beneficial effects of h2SheP expression in a more translationally relevant cardiac tissue set-ting, we compared the conduction properties in monolayers of human-induced pluripotent stem cell-derived cardiomyocytes (hiPSC-CMs) transduced with either control GFP (Supplemen-tary Fig. 5a) or h2SheP (Supplementary Fig. 5b) lentivirus. Similar to NRVMs, transduced h2SheP channels in hiPSC-CMs were successfully trafficked to the cell membrane (Supplementary Fig. 5c) yielding a robust inward current (Supplementary Fig. 5d). Furthermore, optical mapping of AP propagation revealed a

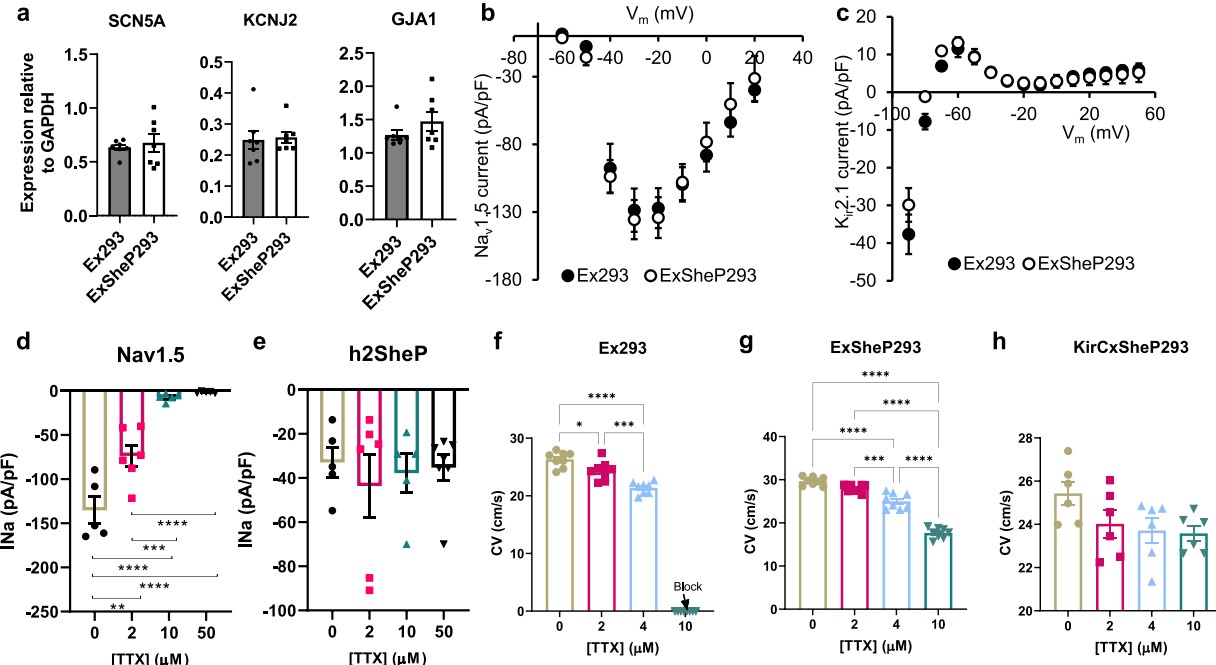

**Fig. 3 Effects of BacNa$_v$ expression in genetically engineered HEK293 cells.** h2SheP was stably expressed in genetically engineered Ex293 line (co-expressing Na$_v$1.5, K$_{ir}$2.1, and Cx43) to create an ExSheP293 line and KirCx293 line (co-expressing K$_{ir}$2.1, and Cx43) to create a KirCxSheP293 line. **a–c** mRNA expression levels of SCN5A, KCNJ2, and GJA1 genes normalized to housekeeping gene GAPDH (**a**, $n = 7$), peak I$_{Nav1.5}$-V (**b**, $n = 5$), or steady-state I$_{K1}$-V (**c**, $n = 5$) curves in Ex293 and ExSheP293 lines showing no effect of h2SheP expression on endogenous channel expression and function. **d**, **e** Increasing concentrations of tetrodotoxin (TTX) led to significant reduction in peak Na$_v$1.5 current in Ex293 cells (**d**, $n = 5$ for 0 and 10 μM groups, $n = 6$ for 2 μM group and $n = 10$ for 50 μM group) but not h2SheP current in KirCxSheP293 cells (**e**, $n = 5$ for 0 and 10 μM groups, $n = 6$ for 2 μM group and $n = 10$ for 50 μM group), showing differential sensitivity of mammalian and prokaryotic Na channels to TTX. All patch-clamp recordings were performed at 25 °C and peak currents of Na$_v$1.5 and h2SheP were measured at −20 and 0 mV, respectively. **P = 0.0011, 0 μM vs. 2 μM group; ***P = 0.0005, 2 μM vs. 10 μM group; ****P < 0.0001, 0 μM vs. 10 μM, 0 μM vs. 50 μM and 2 μM vs. 50 μM groups in **d**. **f–h** Increasing TTX concentrations progressively slowed AP propagation yielding conduction block at 10 μM in Ex293 (**f**, $n = 8$) but not ExSheP293 (**g**, $n = 8$) monolayers, while no CV slowing was observed in KirCxSheP293 monolayers (**h**, $n = 6$). *P = 0.0258, 0 μM vs. 2 μM group; ***P = 0.0006, 2 μM vs. 4 μM group; ****P < 0.0001, 0 μM vs. 4 μM in **f**. ***P = 0.0004, 2 μM vs. 4 μM group; ****P < 0.0001, 0 μM vs. 4 μM, 0 μM vs. 10 μM, 2 μM vs. 10 μM and 4 μM vs. 10 μM groups in **d**. Error bars indicate s.e.m; statistical significance was determined by one-way ANOVA, followed by Tukey's post-hoc test to calculate P-values. Source data are provided as a Source Data file.

1.8-fold higher CV (Supplementary Fig. 5e) in h2SheP- vs. GFP-transduced hiPSC-CM monolayers, and no changes in APD$_{80}$ (Supplementary Fig. 5f) or MCR (Supplementary Fig. 5g).

**Effects of BacNa$_v$ on excitability and conduction in simulated healthy and diseased adult heart tissues.** Since both NRVMs and hiPSC-CMs exhibit immature electrophysiological properties, we next utilized established computational models of adult ventricular myocyte AP and an updated model of the Na$_v$SheP channel[12] (Supplementary Fig. 6) to simulate the effects of h2SheP expression on adult cardiac tissue electrophysiology. Based on similar peak currents recorded from h2SheP and endogenous Na$_v$1.5 channels in NRVMs (Fig. 4b, e), we assigned modeled h2SheP conductance to be at normal (1X) "expression" level when it generated the same peak current as of the endogenous Na$_v$1.5 conductance. We then assessed the effects of different h2SheP conductance levels on properties of the O'Hara-Rudy model of adult human ventricular myocyte AP[25] and found that higher h2SheP expression yielded larger peak Na$^+$ current (Fig. 5a top) and APA (Fig. 5b, c top). This increase in APA, as well as the early plateau phase of the AP, augmented slow delayed rectifier potassium current (I$_{Ks}$, Supplementary Fig. 7c) via an increase in its driving force and channel conductance. The larger I$_{Ks}$ likely opposed the inward BacNa$_v$ current to prevent an

increase in APD$_{80}$ across all simulated h2SheP levels (Fig. 5d top). Importantly, increasing h2SheP current enhanced CM excitability resulting in faster AP upstroke in single cells (Fig. 5e top) and propagation in 1D cable (CV, Fig. 5f top). We further assessed effects of h2SheP expression in a setting of reduced CM excitability[28] where endogenous Na$_v$1.5 current was reduced by 50% (Fig. 5a–f, bottom), which caused no notable change in APA or APD (Fig. 5b–d bottom), decreased AP upstroke velocity (Fig. 5e bottom), and failure of 1D propagation (Fig. 5f bottom). In this simulated "pathological" condition, effects of h2SheP expression were more pronounced, with 1X or a higher expression of h2SheP effectively rescuing both upstroke velocity and CV back to "healthy" levels (Fig. 5e, f bottom). To ensure that the obtained results in simulated human adult CMs are model-independent, we simulated h2SheP expression in two other models of adult ventricular myocyte AP (guinea pig[29,30] and dog[30,31]) and consistently found dose-dependent improvements in tissue excitability and AP conduction due to BacNa$_v$ expression without an increase in APD (Supplementary Fig. 7).

Next, we sought to examine the effects of BacNa$_v$ in models of 2D heterogenous ventricular myocardium in which normally excitable human CMs were separated with randomly distributed nonconducting obstacles akin to collagenous fibrotic tissue[32]. Simulated loss of excitability in 15% of cardiac tissue area decreased average CV from 44 cm/s (for 0% obstacles) to ~34 cm/s

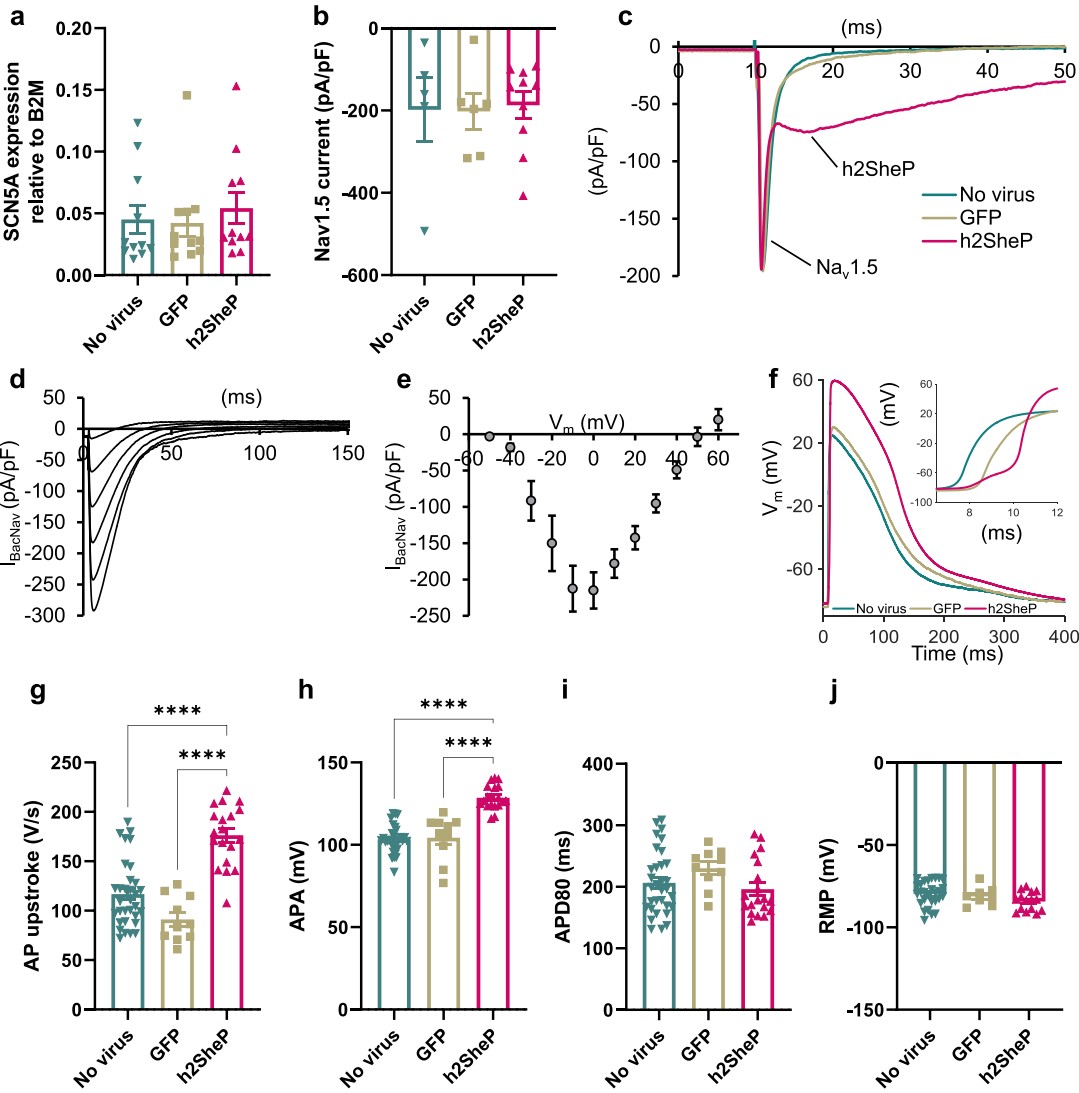

**Fig. 4 BacNa$_v$ expression enhances cardiomyocyte excitability in vitro. a, b** Transduction of NRVMs with MHCK7-GFP ("GFP") or MHCK7-h2SheP-2A-GFP ("h2SheP") lentivirus did not affect mRNA expression of SCN5A gene shown normalized to B2M housekeeping gene (**a**, $n = 11$) or Na$_v$1.5 current density (**b**, $n = 5$ for no virus group; $n = 6$ for GFP group; $n = 10$ for h2SheP group). **c** Representative current responses to a voltage step from $-80$ mV (holding) to $-20$ mV demonstrating slower kinetics of h2SheP than Na$_v$1.5 current. **d, e** Representative h2SheP current responses (**d**) and peak $I_{Na}$–V curve (**e**, $n = 7$) in NRVMs transduced with MHCK7-h2SheP lentivirus. Only traces corresponding to stepping voltages at 0–50 mV are shown in **d**. **f–j** Representative AP traces measured via intracellular recording with inset showing AP upstrokes and corresponding quantifications of maximum AP upstroke velocity (**g**), APA (**h**), APD80 (**i**), and RMP (**j**) in No virus ($n = 32$), GFP ($n = 10$), and h2SheP ($n = 19$) groups. ****$P < 0.0001$ versus h2SheP group in **g** and **h**. Electrophysiological recordings were performed at 25 °C in **b**, **c** and at 37 °C elsewhere. Error bars indicate s.e.m; statistical significance was determined by one-way ANOVA, followed by Tukey's post-hoc test to calculate $P$-values. Source data are provided as a Source Data file.

(Fig. 5g left). Adding h2SheP in this setting accelerated AP conduction in a dose-dependent manner and completely recovered CV with as little as 0.5X h2SheP level (Fig. 5g, h, Supplementary Movie 1). In an independent set of simulations, replacing 20% of cardiac tissue area with random anisotropic nonconducting obstacles, mimetic of interstitial fibrosis, yielded conduction blocks (Fig. 5i, left) at narrow isthmuses with unfavorable current source-sink mismatch[33]. These conduction blocks were successfully overcome by the introduction of h2SheP channels (Fig. 5i right, Supplementary Movie 2).

While reducing endogenous Na current by 50% was used to simulate general loss of CM excitability (Fig. 5a–f), we further explored the therapeutic potential of BacNa$_v$ in a simulated Brugada syndrome model (T1620M mutation in SCN5A) of transmural conduction through endocardial, midmyocardial, and

epicardial CMs (Fig. 5j–o)[34,35]. The rate of fast inactivation of $I_{Na}$ and maximum conductance of transient outward K$^+$ current, $I_{to}$, were varied to model: 1) a "mild" Brugada case with attenuated APA and increased phase 1 notch in the epicardium and midmyocardium, leading to prominent J wave and the "saddle-back" ECG shape (Fig. 5k–n, top) and 2) a "severe" Brugada case resulting in prominent APA attenuation and increase in phase 1 notch in the epicardium and midmyocardium and complete loss of AP dome in the epicardium, leading to "triangular" ECG shape with significant ST elevation (Fig. 5k–n, bottom)[35]. Notably, simulated h2SheP expression in both Brugada settings rescued the changes in the AP amplitude, notch, and dome in a dose-dependent manner (Fig. 5k-m), leading to normalization of the ECG waveforms (Fig. 5n-o). Together, our in silico studies showed that BacNa$_v$ expression in CMs holds the potential to

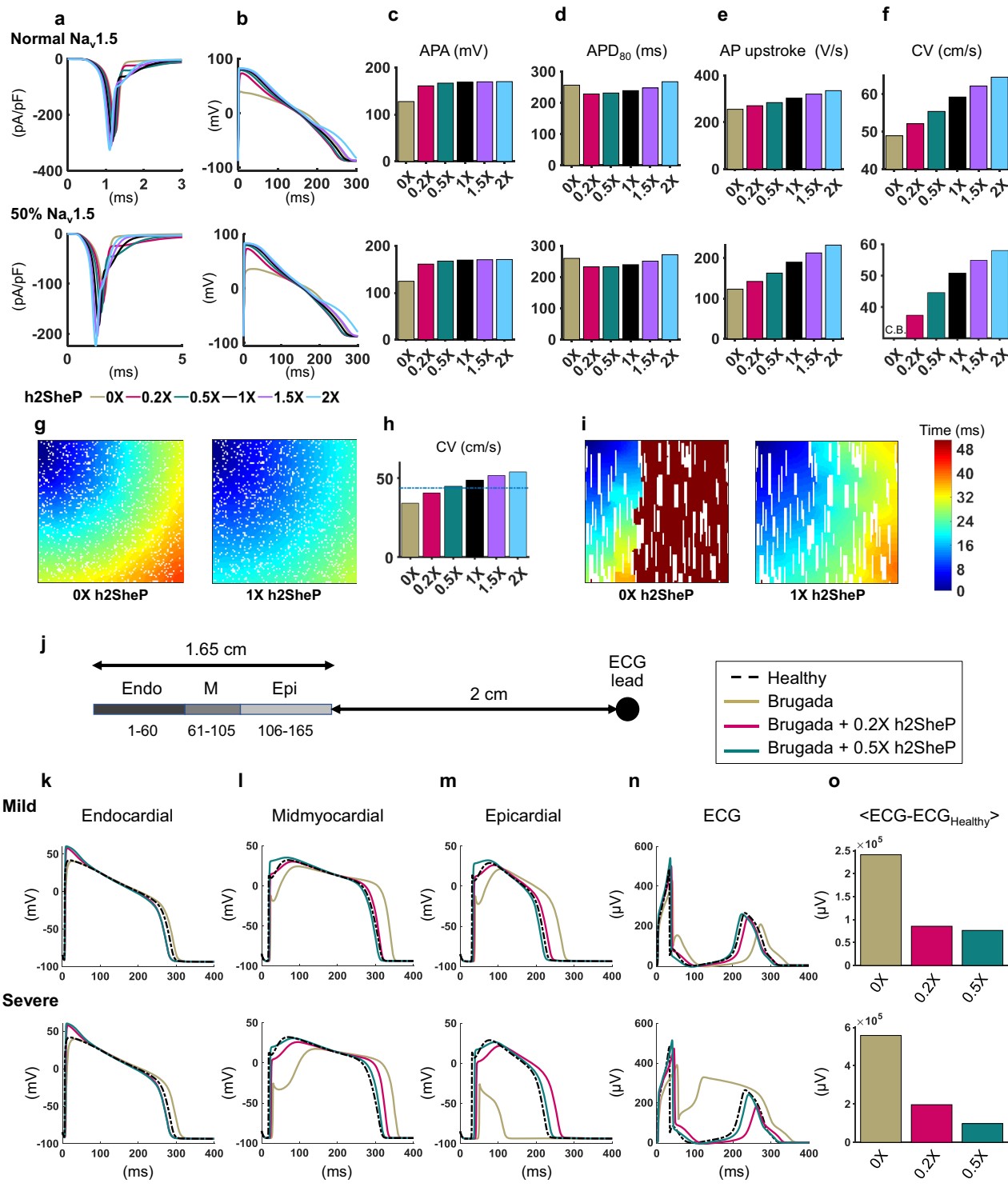

improve impaired AP conduction in the adult heart tissue caused by uniformly (genetically) reduced or heterogeneously lost CM excitability.

**Effects of BacNa_v expression in arrhythmogenic cardiac cell cultures.** Based on our in vitro and in silico studies, we reasoned that expression of BacNa_v in cardiac tissue with slow AP conduction could be antiarrhythmogenic. We thus set to establish an in vitro NRVM monolayer model of the highly arrhythmogenic 2D cardiac substrate with varying fibroblast contents generated

via changes in initial cell seeding density and/or application of an anti-proliferative agent, mitomycin-C. Combinations of lower seeding densities and mitomycin-C treatment generated 2D NRVM tissues with simultaneously reduced CV and APD_80, and high rates (up to 70%) of reentry induction by programmed rapid pacing[36] (Supplementary Fig. 8). We selected a culture condition with the highest reentry incidence and transduced cells with an MHCK7-h2SheP-2A-GFP lentivirus which yielded robust GFP expression in CMs, but not cardiac fibroblasts (Fig. 6a) or untransduced control cells (Fig. 6b). Compared to untransduced and GFP-transduced groups, h2SheP expression in NRVMs led to

**Fig. 5 BacNa$_v$ improves conduction in simulated adult human ventricular tissues and in a model of Brugada syndrome. a** Combined sodium current (from Na$_v$1.5 and added h2SheP) shown during simulated adult human ventricular myocyte AP for normal (top) and reduced (bottom, 50% of normal Na$_v$1.5 current) excitability. Each trace represents a different h2SheP conductance value utilized for the simulation, with 1X representing h2SheP level that produces the same peak current as endogenous Na$_v$1.5 during voltage-clamp simulation. **b–f** Corresponding action potential traces generated with different h2SheP expression levels (**b**) and quantified AP amplitude (APA, **c**), duration (APD$_{80}$, **d**), and maximum upstroke velocity (**e**) modeled in single cells, as well as conduction velocities (CVs) during AP propagation modeled in 1D cables (**f**). Note conduction block (C.B.) in 1D cable with reduced excitability in **f** that is rescued with adding h2SheP. **g**, **h** Isochrone activation maps showing AP conduction in a simulated 1 × 1 cm heterogeneous human ventricular tissue with 15% of the total area (1500 total cells shown in white) being randomly disconnected from the rest of the tissue to model nonconducting obstacles akin to tissue fibrosis and quantified CVs for different levels of added h2SheP (**h**). The obstacle-induced conduction slowing was recovered by the addition of h2SheP (see also Supplementary Movie 1). **i** Activation maps showing AP conduction block without h2SheP (left) and rescued conduction in the presence of 1× h2SheP (right) in a simulated 1 × 1 cm heterogeneous human ventricular tissue with 20% area consisting of nonconducting vertical anisotropic obstacles (shown in white; see also Supplementary Movie 2). In tissue simulations in **g**, **i**, AP conduction was initiated from the top-left tissue corner, with the color bar scale on the far right applying to all activation maps. **j** Schematics describing simulated transmural ventricular AP conduction (60 endocardial, 45 midmyocardial, and 60 epicardial cells; initiated at the endocardial end) and the location of ECG measurement 2 cm from the epicardial surface. **k–o** Simulated AP traces (endocardial, **k**; midmyocardial, **l**; epicardial, **m**), ECG traces (**n**), and corresponding deviations from healthy ECG (**o**) shown for healthy (dashed line) and mild and severe Brugada cases not treated (0×) or treated with h2SheP at 0.2× or 0.5× expression level.

1.4-fold increase in CV (Fig. 6c, d) and APD$_{80}$ (Fig. 6e) and a moderate but not significant decrease in MCR (Fig. 6f). Importantly, in both untransduced (16/27) and GFP-transduced (19/35) monolayers, rapid point pacing caused a conduction block (wave break) distal to the pacing site, followed by the emergence of sustained reentrant waves after pacing was terminated (Fig. 6g, h, Supplementary Fig. 9a, b, Supplementary Movie 3). The h2SheP expression significantly reduced reentry incidence to 22% (4/18 monolayers) by increasing the tissue excitability to prevent the occurrence of distal conduction blocks and instead yielded a partial conduction block proximal to the pacing site without reentry induction (Fig. 6h, Supplementary Fig. 9c, d, Supplementary Movie 3).

To further investigate the potential antiarrhythmic effects of BacNa$_v$ expression in the setting of CM hypertrophy and abnormal Ca$^{2+}$ handling, we treated NRVM monolayers with the α-adrenergic agonist phenylephrine[37] (Supplementary Fig. 10a) and optically mapped AP propagation (Supplementary Fig. 10b). The phenylephrine effects were evident from the increased CM size and APD$_{80}$, while CV was not altered (Supplementary Fig. 10c-e). Importantly, point stimulation in phenylephrine-treated but not untreated monolayers frequently triggered transient focal arrhythmias resulting from delayed afterdepolarizations (Supplementary Movie 4), which were not significantly suppressed with h2SheP treatment (Supplementary Fig. 10f). Since the hypertrophic monolayers had a normal CV, this triggered activity was likely mediated by Ca$^{2+}$-handling rather than a Na$_v$1.5-based mechanism. Together, these in vitro studies demonstrated that h2SheP expression in CMs can improve compromised conduction and decrease the incidence of reentrant arrhythmias in fibrotic cardiac tissues, while potential therapeutic effects on Ca$^{2+}$-handling defects may be limited.

**In vivo AAV-mediated BacNa$_v$ expression in adult mouse heart**. While in vitro transduction of h2SheP in immature CMs resulted in robust expression of functional channels and sodium current (Fig. 4 and Supplementary Fig. 5), potential applications of BacNa$_v$ in vivo would require functional validation of virally delivered h2SheP in adult cardiomyocytes. We thus optimized the dose of self-complementary AAV serotype 9 (scAAV9) vector to obtain global and uniform gene expression throughout the mouse heart six weeks after tail-vein injection (Fig. 7a, b). To investigate the effect of AAV9-mediated h2SheP expression on healthy mouse heart electrophysiology, we measured surface electrocardiograms (ECGs) in mice injected with either scAAV9-MHCK7-h2SheP-HA or scAAV9-MHCK7-GFP control virus 6 weeks post-injection (Supplementary Fig. 11a). We found no

effects of h2SheP on ECG morphology, heart rate, or other measured ECG parameters (PR, QRS, and QTc durations) between the two groups (Supplementary Fig. 11b). In addition, no spontaneous arrhythmias or conduction abnormalities were observed after caffeine and isoproterenol administration, which decreased RR duration similarly in the h2SheP and GFP groups (Supplementary Fig. 11c). These results along with no apparent abnormalities in cardiac structure (Fig. 7a) or mouse behavior 6 weeks after injection, suggested that CM-specific AAV delivery of functional BacNa$_v$ did not adversely affect the healthy heart. Detailed examination of transgene expression was further performed in the sinoatrial node (SAN), which was identified by positive labeling of HCN4 channels and lack of Cx43 expression (Fig. 7c, d). This is of special interest as the exogenous expression of ion channels in the SAN could have deleterious effects on heart rate and physiology. In agreement with the lack of effect on heart rate, GFP or HA labeling was rarely observed in nodal CMs, potentially due to inefficient delivery or expression of the AAV9-MHCK7-driven transgene in the SAN CMs. When immunostained proteins were quantified as the fraction of the F-actin$^+$ tissue area in different heart regions, robust expression of GFP, HA, and Cx43, but not HCN4, was confirmed in the ventricles and atria, while the opposite expression pattern was present in the SAN (Fig. 7e).

To further assess the sarcolemmal expression and distribution of h2SheP at a single-cell level, we administered AAV9-MHCK7-h2SheP-HA-2A-GFP virus to mice and isolated ventricular CMs four weeks later via Langendorff perfusion (Fig. 8a). Immunostaining for the HA tag revealed that the AAV9-delivered channels were targeted to the T-tubular membrane (Fig. 8b, c), known to be rich in endogenous ion channels and transporters[38]. Furthermore, voltage-clamp recordings in dissociated CMs showed the presence of h2SheP current (Fig. 8e) with characteristic peak I-V relationship (Fig. 8f) in cells transduced with h2SheP AAV9 but not in nontransduced cells (Fig. 8d) from the same hearts. At 6 weeks post-AAV injection, we also recorded in current-clamp mode APs from dissociated mouse ventricular CMs (Fig. 8g). While we observed no change in RMP or APA, and a trend towards a higher maximum AP upstroke, the only significant change due to BacNa$_v$ expression in CMs was increased APD$_{90}$ (Fig. 8h–m).

## Discussion

The complex mechanisms underlying electrophysiological disorders in the heart and the growing understanding of their molecular bases make gene therapies a viable treatment option for patients with difficult-to-manage acquired and inherited

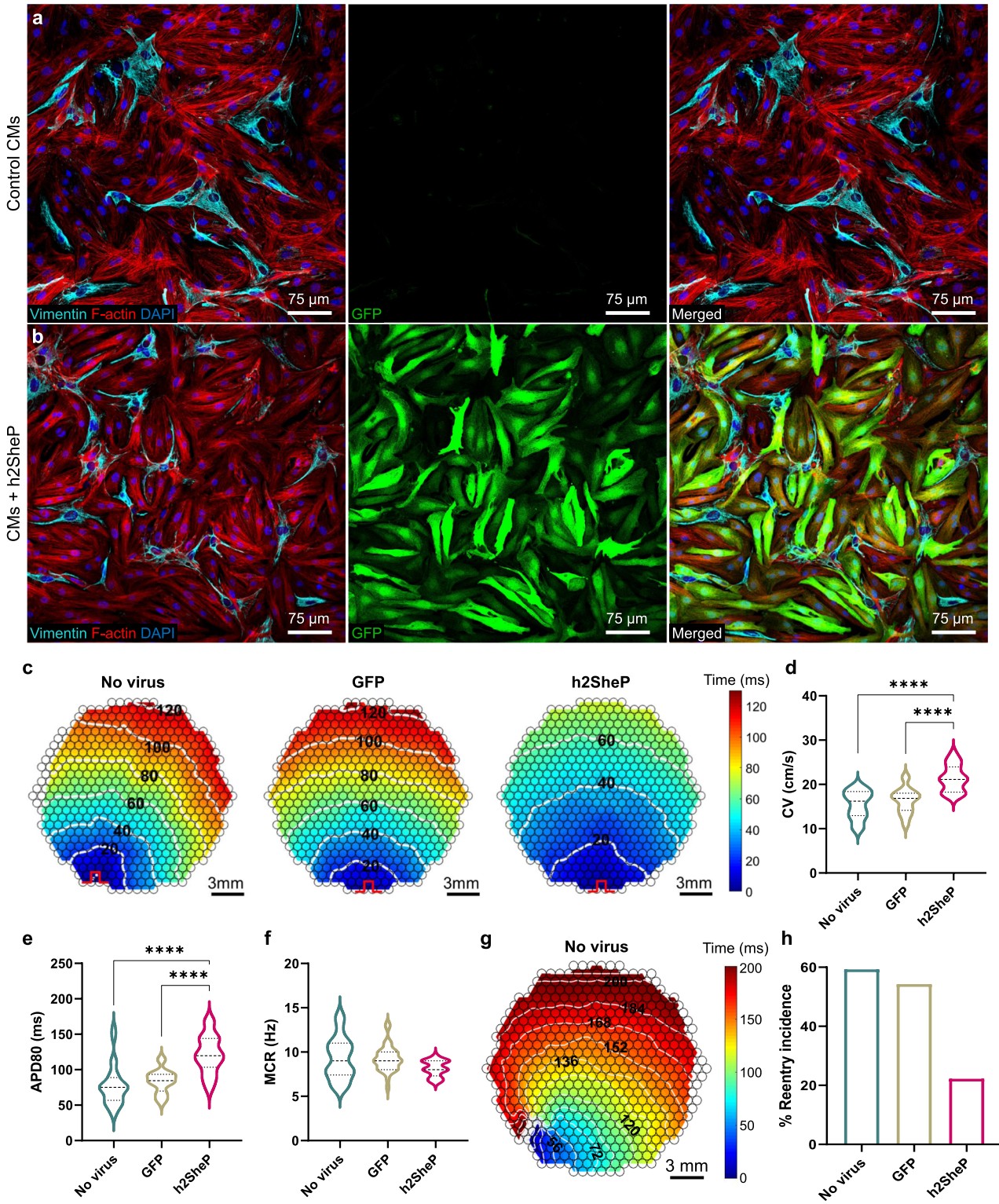

arrhythmias[39–46]. In particular, overexpression of voltage-gated sodium channels could increase cardiac excitability and AP conduction to both decreases the propensity for conduction block and increase the width of the propagating waves. These effects would reduce the incidence of wave breaks and increase the effective tissue size required to sustain reentrant circuits, both of which are known to be antiarrhythmogenic[33,47]. Previous studies have shown that overexpression of the skeletal muscle sodium channel isoform ($Na_v1.4$) in the canine infarct border zone can

increase excitability, improve conduction, and suppress ventricular tachycardia inducibility[48]. However, the observed therapeutic effects were short-lived due to the transient expression profile of adenoviral delivery and the inability to package large mammalian sodium channel genes in an AAV vector for long-term expression. In contrast, the $BacNa_v$ genes are 8–10 times smaller than their mammalian counterparts and thus are not subject to even the stringent size limit of self-complementary (sc) AAVs (~2.3 kb). In this study, we demonstrated stable expression

**Fig. 6 BacNa$_v$ expression improves conduction and prevents reentrant activity in fibrotic cardiomyocyte cultures. a, b** Representative immunostaining images of monolayers containing fibroblasts and NRVMs labelled by vimentin and F-actin, respectively, exhibiting robust cardiac-specific GFP expression in the MHCK-h2SheP-2A-GFP-transduced group (**b**) but not in the nontransduced control (**a**). **c** Representative isochrone activation maps of AP propagation in nontransduced NRVM monolayers ("No virus") and monolayers transduced with MHCK7-GFP ("GFP") or MHCK7-h2SheP-2A-GFP ("h2SheP") lentivirus. **d–f** Monolayers transduced with h2SheP lentivirus ($n = 18$) exhibit improved CV (**d**), longer APD$_{80}$ (**e**), and similar MCR (**f**) compared to nontransduced ($n = 26$) or GFP-transduced ($n = 31$) monolayers. ****$P < 0.0001$ in **d**, **e**. **g** Representative isochrone activation map showing reentrant arrhythmia induced by rapid point pacing in a nontransduced monolayer (see also Supplementary Movie 3). In **c**, **g** pulse signs indicate location of pacing electrode and circles denote 504 recording sites. **h** Transduction with h2SheP lentivirus significantly reduced the rate of reentry incidence (fraction of monolayers with induced reentry) compared to nontransduced and GFP-transduced control groups. Error bars indicate s.e.m; statistical significance was determined by one-way ANOVA, followed by Tukey's post-hoc test to calculate $P$-values. Source data are provided as a Source Data file.

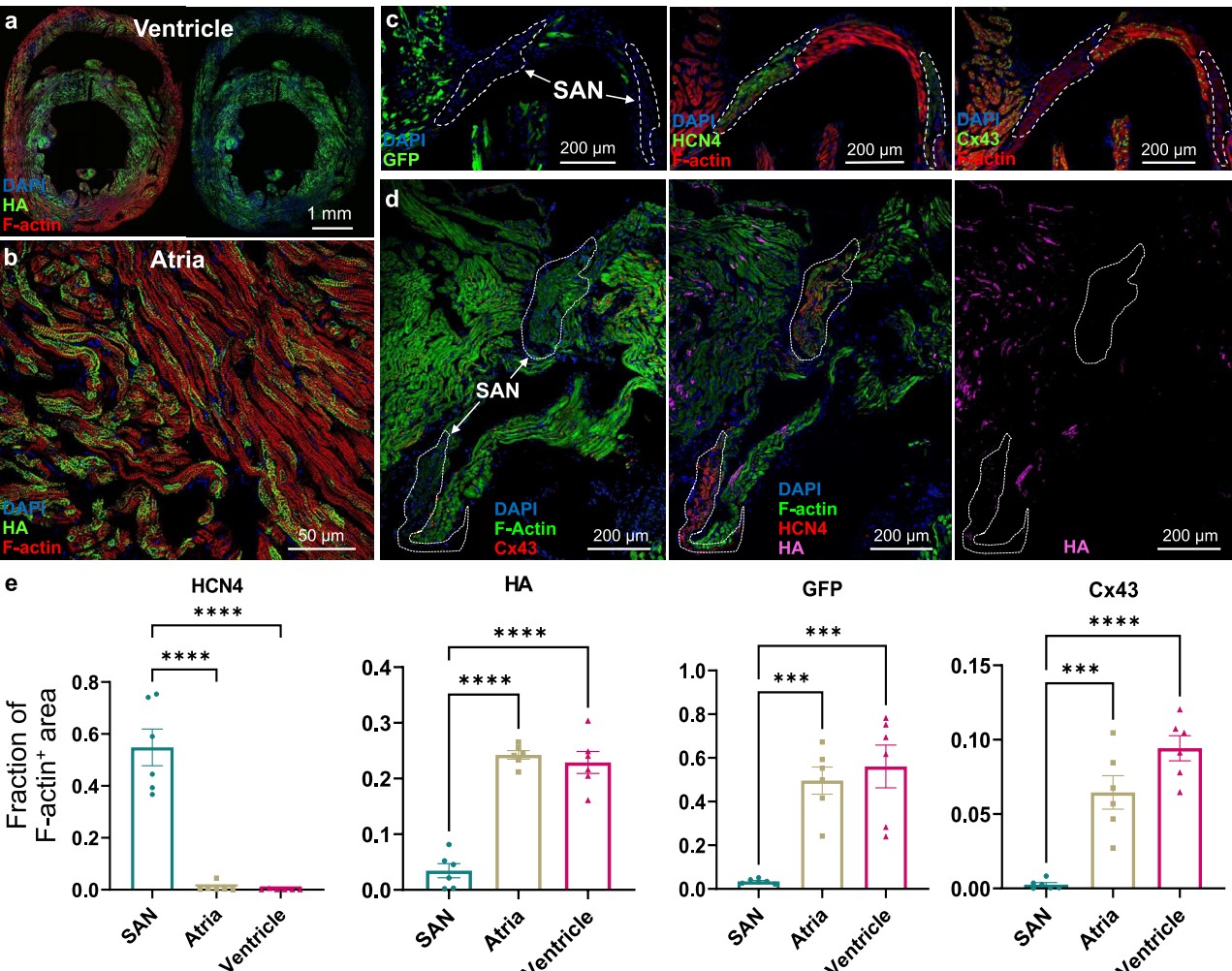

**Fig. 7 Intravenous AAV-mediated delivery of BacNa$_v$ results in robust transgene expression in ventricles and atria, but not the SAN of the adult mouse heart. a, b** Representative images of transverse ventricular (**a**) and atrial (**b**) sections of the mouse heart six weeks after tail-vein injection of $1 \times 10^{12}$ vg of scAAV9-MHCK7-h2SheP-HA showing robust BacNa$_v$ expression in cardiomyocytes. **c, d** Representative images of the sinoatrial node (SAN) and surrounding atria of mice injected with $1 \times 10^{12}$ vg of scAAV9-MHCK7-GFP (**c**) or scAAV9-MHCK7-h2SheP-HA (**d**). The SAN areas are delineated with white dashed lines identified from the robust expression of HCN4 and absence of Cx43. Note minimal transgene expression in the SAN. **e** Quantified areas of HCN4$^+$, HA$^+$, GFP$^+$, or Cx43$^+$ area relative to F-actin$^+$ area in the SAN, atrial, and ventricle ($n = 6$ animals, 6 sections of each tissue per animal were imaged for quantification. ****$P < 0.0001$; ***$P = 0.0006$, SAN vs. Atria in GFP groups; ***$P = 0.0002$, SAN vs. Ventricle in GFP groups; ***$P = 0.0002$, SAN vs. Ventricle in Cx43 groups). Error bars indicate s.e.m; statistical significance in **e** was determined by one-way ANOVA, followed by Tukey's post-hoc test to calculate $P$-values. Source data are provided as a Source Data file.

of functional BacNa$_v$ channels in neonatal rats, human iPSC-derived, and adult mouse CMs. The BacNa$_v$ expression improved excitability and AP conduction in rat and human CMs in vitro (Fig. 4, Supplementary Fig. 5) and in simulated healthy and diseased adult cardiac tissues from multiple species in silico (Fig. 5, Supplementary Fig. 7), and effectively reduced incidence of reentrant activity in fibrotic neonatal rat cardiac cultures (Fig. 6). Together, these results establish a foundation for the potential use of the BacNa$_v$ platform as a gene-based therapy for cardiac conduction disorders.

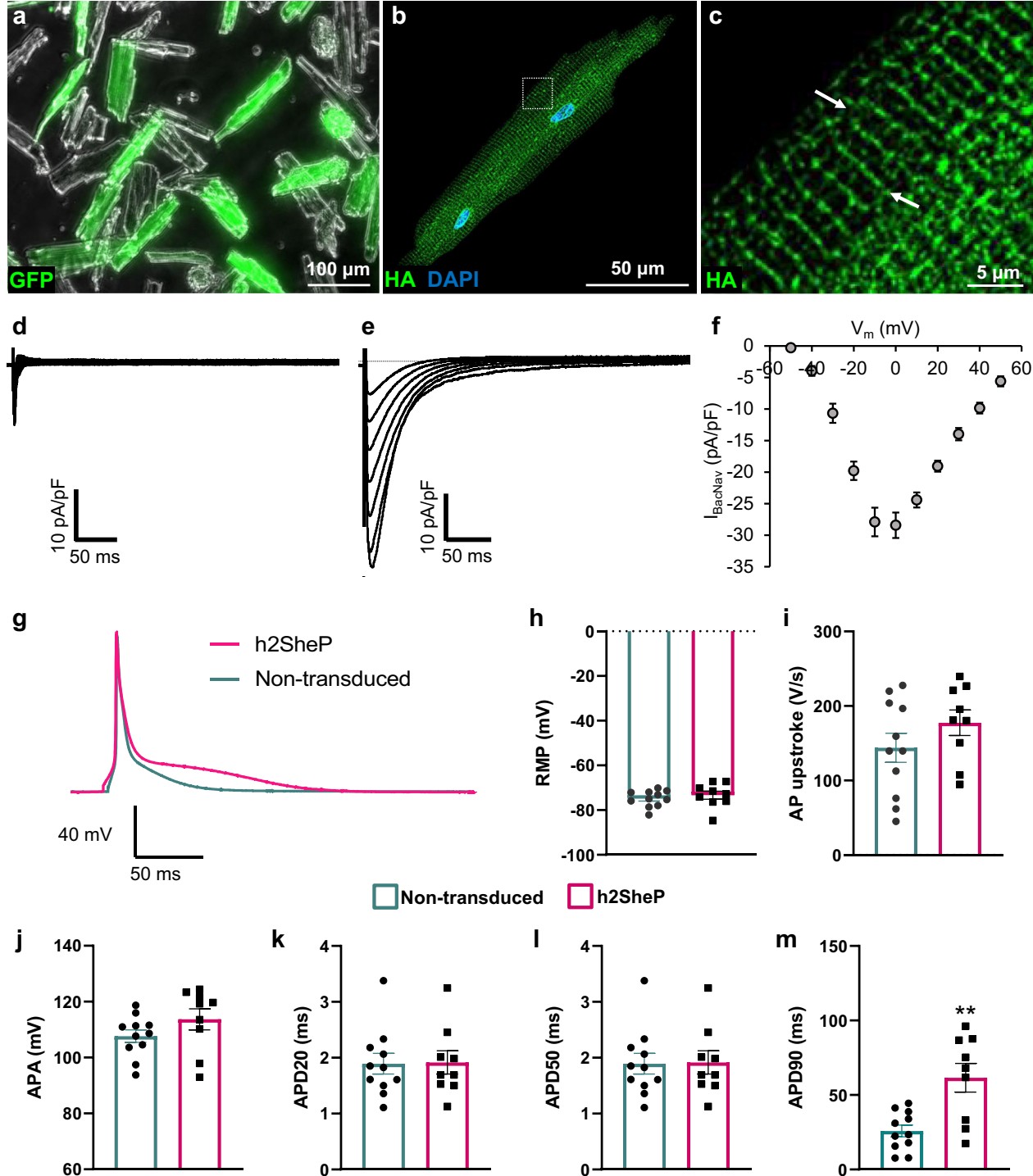

**Fig. 8 Intravenous AAV-mediated delivery of BacNa$_v$ yields expression of functional channels in mouse ventricular myocytes. a–c** Representative images of dissociated cardiomyocytes (CMs) (**a**) from mouse ventricles four weeks after tail-vein injection with $1 \times 10^{12}$ vg of AAV9-MHCK7-h2SheP-HA-2A-GFP showing expression of h2SheP-HA channels at T tubules (**b**, **c**, examples shown with white arrows). **d**, **e** Representative sodium current traces in response to voltage steps from −80 mV (holding potential) to test potentials from −50 to 50 mV recorded from nontransduced (**d**) and transduced, h2SheP-expressing (**e**) mouse ventricular myocytes four weeks after tail-vein injection of $2 \times 10^{12}$ vg of AAV9-CAG-h2SheP-2A-GFP. **f** Corresponding peak $I_{Na}$–V curve for CMs transduced with h2SheP virus ($n = 5$). Patch-clamp recordings in **d**–**f** were performed in the presence of 50 µM TTX. **g**–**m** Representative action potential (AP) traces recorded from nontransduced and transduced h2SheP-expressing mouse ventricular myocytes six weeks after tail-vein injection with $1 \times 10^{12}$ vg of AAV9-MHCK7-h2SheP-HA-2A-GFP (**g**) and corresponding resting membrane potential (RMP, **h**), maximum upstroke velocity (AP upstroke, **i**), AP amplitude (APA, **j**), and durations (APD$_{20}$, **k**; APD$_{50}$, **l**; APD$_{90}$, **m**, **\*\***$P = 0.0017$). $n = 9$ for nontransduced and $n = 11$ for transduced CMs. All patch-clamp recordings were performed at 25 °C. Error bars indicate s.e.m; statistical significance in **m** was determined by unpaired two-tailed $t$-test. Source data are provided as a Source Data file.

In our study, we achieved significant improvement in BacNa$_v$ expression level in cardiomyocytes via codon optimization and promoter selection. In recent years, codon optimization has become a very effective tool to improve the mammalian expression of microorganism-derived genes for various applications, including genome editing[49] and optogenetics[50]. In our study, codon-optimized BacNa$_v$ sequences showed 3–5-fold higher current density compared to wild-type sequences (Fig. 1), suggesting codon optimization as the first step in the future development of therapeutic BacNa$_v$ orthologs. Further increase in BacNa$_v$ expression could be achieved by the use of strong promoters (Fig. 2); however, other factors, such as the need for CM-specific expression, must also be considered in selecting the optimal promoter for in vivo applications. Several naturally derived and synthetic promoters with strong CM specificity have been developed[51–53] including the MHCK7[17] and cTnT[16] promoters used in this study. In addition, the atrial natriuretic factor (ANF) promoter[54] has been shown to largely restrict gene expression to atrial myocytes, making it suitable for targeting atrial arrhythmias including fibrillation[55,56]. While the use of tissue-targeted engineered AAV capsids[57] and CM-specific promoters could prevent nonspecific expression, in the case of BacNa$_v$, the requirement for significantly hyperpolarized resting membrane potential would limit any off-target channel activity to excitable tissues only. However, unwanted gene expression in the sinoatrial node could still adversely affect cardiac pacemaking. This study for the first time examined the expression of an intravenously AAV9-delievered transgene in the SAN and found minimal nodal expression (Fig. 7), potentially resulting from low AAV9 entry or MHCK7 promoter activity in the SAN CMs. In fact, compared to atria, the mouse SAN expresses less laminin receptor LamR (*Rpsa*)[58], important for AAV9 entry[59], as well as an alpha-myosin heavy chain (*Myh6*) and muscle creatine kinase (*Ckm*) whose enhancer/promoter regions comprise the MHCK7 regulatory cassette[17]. For the future cardiac gene therapies that target the cardiac conduction system, it will be critical to systematically explore transduction efficiency with other AAV serotypes, promoters, and/or delivery routes including intracoronary and intramuscular injection.

With the goal of maximizing the physiological expression of BacNa$_v$ in CMs, it is also important to ensure that the channels are efficiently trafficked to intended membrane compartments. In this study, we examined BacNa$_v$ trafficking in immature and adult CMs by introducing an HA tag at the channel C terminus. While BacNa$_v$ appeared efficiently targeted to the plasma membrane in NRVMs and hiPSC-CMs (Supplementary Fig. 5c), it primarily localized at the T-tubular compartment of adult CMs (Fig. 8b, c). In contrast, Na$_v$1.5 channels in adult CMs localize to the intercalated disks, lateral membranes, and T tubules[60–63] where they play distinct roles in CM excitability, slow conduction, and excitation-contraction coupling[64–66]. It is possible that the inclusion of PDZ-[67] or Ankyrin G-[68] binding motifs could help additionally direct BacNa$_v$ to the lateral membrane and intercalated disk and enhance their potential for improving compromised cardiac conduction.

In our study, h2SheP expression in CMs significantly improved CV and decreased incidence of wave breaks and reentrant arrhythmias in fibrotic NRVM monolayers in vitro (Fig. 6, Supplementary Fig. 9) and rescued slow conduction and conduction block in simulated adult cardiac tissues from multiple species in silico (Fig. 5, Supplementary Fig. 7). This has been the direct result of enhanced CM excitability due to gain of peak sodium current (Fig. 5a and Supplementary Fig. 7a, b), evident from the increased AP upstroke and amplitude without the change in resting membrane potential (Fig. 4g–j, Fig. 5a and Supplementary Fig. 7a, b). While a large class of arrhythmias is precipitated by

reduced excitability and fibrosis, others involve acquired and congenital abnormalities in Ca$^{2+}$ and K$^+$ channels or various signaling molecules[69–71], where BacNa$_v$ expression may not be a therapy of choice. Indeed, in the setting of phenylephrine-treated hypertrophic NRVMs, the incidence of triggered focal arrhythmias resulting from defects in Ca$^{2+}$-handling[37,72–74] was not affected by h2SheP expression (Supplementary Fig. 10). Importantly, in none of our in vitro, in silico, or in vivo studies, have we found any evidence for h2SheP-induced cardiac arrhythmias.

BacNa$_v$ effects on cardiac AP are also expected to be species-specific. The extra inward current from h2SheP expression in our studies did not prolong APD in CMs with relatively long APs (>100 ms), such as serum-cultured NRVMs (Figs. 2f, 4i, and Supplementary Fig. 10d), human iPSC-CMs (Supplementary Fig. 5f), or simulated guinea pig, dog, or human ventricular CMs (Fig. 5b, d, Supplementary Fig. 6a, b). This is likely due to the h2SheP-induced increase in AP peak and early repolarization potentials (Figs. 4h, 5c) that led to an increase in I$_{Ks}$ (Supplementary Fig. 6c). In contrast, in cells with short APs (<100 ms), such as Ex293 (Supplementary Fig. 2b, c), serum-free cultured NRVMs (Fig. 6e), and adult mouse CMs (Fig. 8m), h2SheP current yielded APD prolongation likely due to low I$_{Ks}$ expression in these cells[75–77]. Moreover, in h2SheP-expressing mouse CMs, no increase in APA, APD$_{20}$, or APD$_{50}$ (Fig. 8j–l) along with APD$_{90}$ prolongation, suggested that the relatively large transient outward K$^+$ current (I$_{to}$)[78–80] additionally opposed h2SheP current to prevent an increase in repolarizing K$^+$ currents.

Overall, due to their fast heart rate and short AP, mice do not appear to be a suitable model to evaluate the therapeutic efficacy of BacNa$_v$ in vivo. In fact, h2SheP channels are largely inactive at high mouse resting heart rates (~10Hz)[12], which was evident from no QT prolongation found in surface ECGs (Supplementary Fig. 11b), despite the observed APD$_{90}$ prolongation in isolated CMs. While our findings suggest that BacNa$_v$ expression in human hearts (which have relatively long APD) would not be arrhythmogenic, studies in larger animal models with human-like cardiac pathophysiology will be needed to further investigate the therapeutic potential of BacNa$_v$. Additionally, considering the known roles of APD dispersion in arrhythmia induction[81,82], future studies should also involve species-specific modeling of how BacNa$_v$ kinetics affects the electrophysiology of different myocardial layers (Fig. 5j–o). The experimental-computational platform described in this study provides a blueprint for accomplishing this goal.

In summary, our studies demonstrate that prokaryotic sodium channels can be directly, specifically, and stably expressed in cardiomyocytes through viral gene delivery to augment tissue excitability and conduction. These findings warrant further development of antiarrhythmic BacNa$_v$ gene therapies in large animal models of disease for potential clinical translation.

## Methods

All animal studies were performed in accordance with the animal protocol A064-21-03 approved by the Duke University Institutional Animal Care and Use Committee.

**Plasmid construction**. All lentiviral transfer plasmids were constructed from the pRRL-CMV vector (a gift from Inder Verma, Salk Institute). Human codon optimization of bacterial Na$_v$SheP D60A[7,12] (bSheP) gene was performed via Genscript OptimumGene algorithm[83] (hSheP) and ATUM Gene-GPS™ algorithm[84] (h2SheP). Wild-type and codon-optimized sequences are listed in Supplementary Information. Human codon-optimized cDNAs synthesized by respective companies were subcloned into the pRRL-CMV vector where they were linked with GFP via the T2A peptide (pRRL-CMV-hSheP-2A-GFP and pRRL-CMV-h2SheP-2A-GFP). The lentiviral plasmid containing wild-type channel sequence co-expressed with GFP (pRRL-CMV-bSheP-2A-GFP) served as the control. For optimization of transcription efficiency in cardiomyocytes, two additional lentiviral transfer plasmids were constructed by replacing the CMV

promoter in pRRL-CMV-h2SheP-GFP with MHCK7[17] and cTnT[16] promoters (pRRL-MHCK7-h2SheP-2A-GFP and pRRL-cTnT-h2SheP-2A-GFP). Single-stranded and self-complementary AAV transfer plasmids were constructed from the pAAV-CAG-eYFP (a gift from Viviana Gradinaru, Addgene plasmid #104055) and pscAAV-CAG-GFP (a gift from Mark Kay, Addgene plasmid #83279), respectively. For mouse tail-vein injection studies, h2SheP-2A-GFP and MHCK7-h2SheP-HA fragments were amplified from lentiviral plasmids and subcloned into AAV vectors to generate pAAV-MHCK7-h2SheP-HA-2A-GFP, pAAV-MHCK7-GFP, pAAV-CAG-h2SheP-2A-GFP, and pscAAV-MHCK7-h2SheP-HA. Plasmid pscAAV-CAG-GFP[85] (a gift from Mark Kay, Addgene plasmid #83279) was used to generate control scAAV9 for optimization of in vivo delivery method.

**Flow cytometry.** HEK293 (ATCC, CRL-1573) monolayers were rinsed with phosphate-buffered saline (PBS) then dissociated using 0.05% Trypsin-EDTA (Thermo Fisher Scientific) at 37 °C for 3 min. Trypsin was quenched with DMEM high glucose (Thermo Fisher Scientific) containing 10% FBS (Hyclone) and 20 μg/ml DNase I (Millipore 260913). The cell suspension was centrifuged at $500 \times g$ for 5 min, then resuspended in 4% paraformaldehyde (PFA) diluted in PBS. Cells were incubated in 4% paraformaldehyde (PFA) for 10 min at room temperature (RT), centrifuged again, then resuspended in PBS containing fluorescence-activated cell sorting (FACS) buffer (PBS with 0.5% BSA (Sigma), 0.1% Triton-X 100 (Thermo Fisher Scientific), and 0.02% Azide (VWR)). FACS was performed using either BD DiVA or B-C Astrios cell sorter at the Flow Cytometry Shared Resource Core Facility at Duke University. The analysis was performed using FlowJo v10.7.1.

**Lentivirus production and titration.** High-titer lentiviruses were prepared using second-generation lentiviral packaging system as described previously[13]. Specifically, 293T cells (ATCC, CRL-3216) were co-transfected with lentiviral transfer plasmid, packaging plasmid psPAX2, and envelope plasmid pMD2.G (6:3:1 mass ratios) using JetPRIME transfection reagent (Polyplus). Seventy-two hours after transfection, the supernatant containing lentiviral particles was collected, centrifuged ($500 \times g$, 10 min), and filtered through 0.45 μm cellulose acetate filter (Corning) to remove cell debris before being combined with Lenti-X Concentrator (Clontech) at 3:1 volume ratio and incubated overnight at 4 °C. Concentrated lentiviral particles were harvested following 45 min centrifugation ($1500 \times g$, 4 °C) and resuspended in DPBS. Plasmids psPAX2 and pMD2.G were obtained from Didier Trono (Addgene plasmids #12260 and #12259). To determine the functional titer of lentiviruses expressing fluorescence reporter, 293T cells were transduced with serial dilutions of concentrated lentiviral stock and the percentage of transduced cells was determined via flow cytometry 72 h post-transduction. Functional titer in transduction units per mL (TU/mL) was estimated from dilutions that yielded 5–30% transduction efficiency, by dividing the total number of transduced cells by the volume of virus added in mL. Transduction in HEK293, NRVM, and hiPSC-CM monolayers was performed with the multiplicity of infection (MOI) of 1, 7, and 2, respectively, and functional studies were conducted 3–5 days after transduction.

**Neonatal rat ventricular myocyte culture.** Ventricles of both male and female 2-day-old Sprague-Dawley rats (Charles River Laboratories, Wilmington MA) were excised, minced, and incubated with 0.1% trypsin (Thermo Fisher Scientific) overnight and dissociated in four sequential steps using 0.1% collagenase[36]. Dissociated cells were centrifuged for 5 min at $200 \times g$ and further enriched by a 45 min preplating step. Isolated cardiomyocytes were seeded onto Aclar coverslips (21 mm diameter, Electron Microscopy Sciences) coated with 30 μg/ml fibronectin (Sigma) at $8 \times 10^4$ cells/cm² in DMEM/F12 medium (Gibco, 11320-033) supplemented with 10% fetal bovine serum (FBS), 0.2% penicillin, and 0.2% B12. The following day (day 1), cells were treated with 10 μg/ml mitomycin-C (Sigma) for 2 h before media change to fresh seeding media. At day 2, media was changed to serum-free maintenance media (DMEM/F12 + 0.2% penicillin + 0.2% B12 + 2.5 μg/ml L-ascorbic acid + 5 nM Triiodo-L-Thyronine + 1X Insulin-Transferrin-Selenium supplement) and lentivirus was added into the cultures. Complete maintenance media change was performed every 2 days and cultures were studied on days 4–6. For optimization of highly arrhythmogenic cultures, four cell seeding numbers were tested—300K, 400K, 600K, and 800K, which correspond to seeding densities of $8 \times 10^4$, $1.1 \times 10^4$, $1.6 \times 10^5$, and $2.2 \times 10^5$ cells/cm², respectively. For patch-clamp and sharp electrode recording studies, maintenance media consisted of DMEM/F12, 0.2% penicillin, 0.2% B12, and 5% FBS. For experiments in hypertrophic NRVM monolayers, 100 μM phenylephrine (Sigma–Aldrich) was added for 24 h at Day 3 and Day 8 of culture and monolayers were optically mapped and immunostained at culture Day 9[37].

**hiPSC-CM differentiation and culture.** Human-induced pluripotent stem cells (hiPSCs) were reprogrammed from commercially available BJ fibroblasts (ATCC cell line, CRL-2522) at the Duke University iPSC Core Facility and named DU11 (Duke University clone #11)[86]. The DU11 hiPSC line was authenticated by pluripotency marker expression using IF and FACS, karotyping to confirm genomic integrity, and teratoma formation[86]. DU11 hiPSCs were differentiated into cardiomyocytes (hiPSC-CMs) using small-molecule modulation of the Wnt pathway[87] and purified via metabolic selection[88] on day 10 post induction[86]. Specifically,

DU11 hiPSCs were plated at $2 \times 10^5$/cm² with 5 μM Y-27632 (ROCK inhibitor, Tocris) and induced to differentiate 2 days after seeding. To induce cardiac differentiation (on day 0, d0), cells were treated with 10–14 μM CHIR99021 (SelleckChem) in RPMI-1640 with B27(−) insulin (ThermoFisher Scientific). Exactly 24 h later, CHIR was removed and replaced with basal RPMI/B27(−) medium. On d3, half of the old medium was collected and mixed with fresh RPMI/B27(−) medium containing 5 μM (final concentration) IWP-4 (Tocris). On d5, IWP-4 was replaced with a basal RPMI/B27(−) medium. From d7 onward, cells were fed with RPMI/B27(+)-insulin every 2–3 days, with spontaneous beating generally starting on d7–d10 of differentiation. Differentiating CM cultures were purified via metabolic selection between d10 and d12 by rinsing cultures with PBS, followed by incubation in "no glucose" medium for 48 h (glucose-free RPMI (ThermoFisher Scientific 11879020) supplemented with 4 mM lactate (Sigma L4263), 0.5 mg/mL recombinant human albumin (Sigma A6612), and 213 μg/mL L-ascorbic acid 2-phosphate (Sigma A8960))[88]. Metabolically purified hiPSC-CMs were dissociated into single cells and plated onto 21 mm diameter Aclar coverslips coated with Corning Matrigel hESC-Qualified Matrix (Corning, 354277) at $2 \times 10^5$ cells/cm² (for optical mapping) or $10^4$ cells/cm² (for patch clamp) in 3D RB+ medium, which contains RPMI-1640 (Sigma, R8758), 2% B27 supplement (Gibco, 17504044), 2 mg/mL aminocaproic acid (Sigma, A2504), 50 μg/mL ascorbic acid 2-phosphate (Sigma, A8960), 1% penicillin-streptomycin (Thermo Fisher,15140), 1% non-essential amino acids (Thermo Fisher, 11140), 1% sodium pyruvate (Thermo Fisher, 11360), 0.45 μM 1-thioglycerol (Sigma, M6145), and 5μM Y-27632 (Tocris, 1254). One day post-seeding, the medium was replaced with 3D RB+ medium without Y-27632 (maintenance medium) and h2SheP or control lentivirus was added. The medium was exchanged every other day and cells underwent patch-clamp or optical mapping 72–96 h after lentiviral transduction.

**Quantitative RT-PCR.** Total RNA was extracted using RNeasy Plus Mini Kit according to the manufacturer's instructions (Qiagen) and the concentration was measured using NanoDrop One (Thermo Scientific). Reverse transcription was run on equal amounts of RNA using iScript cDNA Synthesis Kit (Bio-Rad). Standard quantitative PCR was performed using an iTaq Universal SYBR Green Supermix kit (Bio-Rad). The relative expression of indicated genes was quantified by the ΔCT method[89]. The primers used are listed in Supplementary Table 1.

**Whole-cell patch-clamp recordings.** Dissociated single cells were plated onto Aclar coverslip and left to attach for 5 h in 37 °C incubators. Coverslip was then transferred to a glass-bottom patch-clamp chamber perfused with bath solution. Patch pipettes were fabricated with tip resistances of 1–2 MΩ when filled with pipette solution. Whole-cell patch-clamp recordings were acquired at room temperature (25 °C) or 37 °C using the Multiclamp 700B amplifier (Axon Instruments), filtered with a 10 kHz Bessel filter, digitized at 40 kHz, and analyzed using WinWCP software (John Dempster, University of Strathclyde). To measure activation properties of voltage-gated sodium channels, membrane voltage was stepped from a holding potential of -80 mV to varying 500 ms test potentials (−50 to 60 mV, increments of 10 mV). Inactivation of voltage-gated sodium channels was derived from peak currents measured at 0 mV after varying 3-s prepulse potentials (−160 to −30 mV, increments of 10 mV). Steady-state $I_{K1}$–V curve was generated from the current responses to varying 1 s test potentials (−90 to 50 mV, increments of 10 mV) from a holding potential of −40 mV. Action potentials were triggered by injecting a 1 ms current pulse at 1.1× threshold amplitude. For $I_{K1}$ and AP recordings, Tyrode's solution was used as bath solution, containing (in mM): 135 NaCl, 5.4 KCl, 1.8 CaCl₂, 1 MgCl₂, 0.33 NaH₂PO₄, 5 HEPES, and 5 glucose; and pipette solution containing (in mM): 140 KCl, 10 NaCl, 1 CaCl₂, 2 MgCl₂, 10 EGTA, 10 HEPES, and 5 MgATP. For sodium current recordings, bath solution consists of (in mM): 135 NaCl, 1.8 CaCl₂, 1.2 MgCl₂, 2 NiCl₂, 10 HEPES, and 10 glucose; pipette solution consists of (in mM): 115 CsCl, 10 NaCl, 0.5 MgCl₂, 10 TEA-Cl, 10 EGTA, 10 HEPES, and 5 MgATP. Tetrodotoxin (TTX) in micromolar concentrations was also included in bath solution in studies where blockade of $Na_v1.5$ was desired.

**Sharp intracellular recordings.** Coverslip plated with confluent NRVM monolayer was transferred into a patch-clamp chamber perfused with Tyrode's solution at 37 °C. The cell monolayer was paced at 1 Hz by a bipolar point electrode and propagated APs at cells remotely situated from the stimulus site were recorded with a high-access resistance electrode (50–100 MΩ) filled with 3M KCl. Data were acquired and processed in similar manners to whole-cell current-clamp recordings. AP parameters, including resting membrane potential (RMP), AP amplitude (APA), AP duration at 80% repolarization (APD₈₀), and maximum AP upstroke velocity, were extracted using a custom-made MATLAB script.

**Optical mapping of action potential propagation and reentry induction in cardiomyocyte cultures.** Confluent cell monolayers were optically mapped with a 20 mm diameter hexagonal array of 504 optical fibers (Redshirt Imaging), as previously described[18,36,90]. Specifically, monolayers were stained with 10 μM Di-4-ANEPPS (Biotium, 61010) for 5 min at room temperature before being transferred to a temperature-controlled (37 °C) recording chamber filled with Tyrode's solution. Illumination via a solid-state excitation light source (Lumencor, SOLA SM)

was passed through a 520 ± 30 nm bandpass filter to excite the dye, and emitted red fluorescence signals (λ > 590 nm) were collected by the optical fiber array, converted to voltage signals by photodiodes, and recorded at a 2.4 kHz sampling rate with a 750 μm spatial resolution. Action potential propagation was initiated by 10 ms, 1.2 × threshold, 1 Hz stimuli from a bipolar point electrode connected to a Grass Stimulator (Grass Technologies). Light shutter control, data acquisition, and electrical stimulation were synchronized using LabView 8.5. Maximum capture rate (MCR) was determined as the highest pacing rate at which tissue responded in 1:1 fashion. Generation of isochrone maps and calculation of CV and $APD_{80}$ were performed for 1 Hz pacing using custom MATLAB software, as previously described[91,92]. For reentry induction[93], NRVM monolayers were stimulated with 15 pulses, at the maximum 1:1 capture rate (MCR). If reentry was not induced and 1:1 capture during pacing was maintained, the pacing rate was increased by 0.5 Hz in the next induction attempt. If 1:1 capture during pacing was lost, the rate was decreased by 0.25 Hz and the monolayer stimulated again. The resulting success or loss of 1:1 capture was then followed by an increase or decrease of pacing rate by 0.125 Hz, respectively, as the last attempt at induction. In the case of successful reentry induction, the recording was performed 1, 2, and 5 min later to assess if reentry was sustained long term. Incidence of reentry induction was calculated as the fraction of total monolayers in which sustained reentry (>1 min) was successfully induced.

**Computational modeling.** $BacNa_v$ model was adapted from Nguyen et al.[12] with updated voltage-clamp and current-clamp experimental data. Modifications were made to the time constant ($\tau_m$ and $\tau_h$) and steady-state functions for activation and inactivation ($m_\infty$ and $h_\infty$) as follows:

$$\tau_m = \frac{34.65}{\exp\left(\frac{V_m + 43.47}{14.36}\right) + \exp\left(-\frac{V_m + 15.75}{0.2351}\right)} + 1.66 \quad (1)$$

$$\tau_h = \frac{107.8}{\exp\left(\frac{V_m + 27.15}{0.1281}\right) + \exp\left(-\frac{V_m + 25.63}{25.19}\right)} + 9.593 \quad (2)$$

$$m_\infty = \frac{1}{1 + \exp\left(\frac{-22.5 - V_m}{2.704}\right)} \quad (3)$$

$$h_\infty = \frac{1}{1 + \exp\left(\frac{V_m + 77.05}{10.64}\right)} \quad (4)$$

The new form of time constant functions was chosen for its ability to produce a wide variety of curve shapes including Gaussian distribution but with asymmetry defined by the shape-fitting parameters. Modeling of $BacNa_v$ effects in different adult cardiomyocyte models was achieved by inserting the $BacNa_v$ equations directly into the Rudy lab models of human[25], dog[30,31], and guinea pig[29,30] ventricular myocyte. One-dimensional (1D) cable simulations of AP propagation were performed as described previously[12] using 100 μm cell length, 10 μm cell radius, 1 cm total cable length (100 total cells), and 0.4 kΩ.cm intracellular resistivities. All of human, dog, and guinea pig models were paced at their respective sinus rhythm rates (1 Hz for human, 2 Hz for dog, and 3.33 Hz for guinea pig) until reaching equilibrium (defined when all state variables had variability of <0.001%/beat) and parameters of the last induced AP (in the single-cell model; AP upstroke, APA, and $APD_{80}$) or conducted AP (in the 1D cable; CV) were determined and used for comparisons among different conditions. Two-dimensional (2D) human cardiac tissue simulations were implemented as a continuous monodomain model. Nonconducting obstacles were randomly generated using a custom MATLAB GUI. After the locations of obstacle nodes were determined, the conductivity was set to 0 for the connections from obstacle nodes to all other nodes and vice versa. Domains were discretized into 100 patches by 100 patches with dx=dy=0.01 cm for a total domain dimension of 1 cm by 1 cm. Human cardiomyocyte formulation and all other conduction parameters (e.g., intracellular resistivity) were maintained from the 1D cable model. For Brugada syndrome simulations, transmural conduction in a cable of 165 guinea pig ventricular myocytes was modeled as previously described[35]. Specifically, the cable was divided into the endocardial (cells 1–60), midmyocardial (cells 61–105), and epicardial (cells 106–165) region and stimulated at the endocardial end. The three regions were differentiated by the density of the transient outward potassium current ($I_{to}$) and the ratio of current density between the slow and rapid rectifying potassium currents ($I_{Ks}:I_{Kr}$). Endocardial cells had zero $I_{to}$ and an 11:1 ratio of $I_{Ks}:I_{Kr}$; midmyocardial cells had a max $I_{to}$ of 0.2125 pA/pF and a 4:1 ratio of $I_{Ks}:I_{Kr}$; and epicardial cells had an $I_{to}$ of 0.25 pA/pF and a 35:1 ratio of $I_{Ks}:I_{Kr}$. Brugada severity was simulated at two levels by increasing both $I_{to}$ maximum conductance and the speed of fast inactivation for the endogenous $Na_v1.5$ current as previously described[34]. Specifically, mild Brugada case was modeled with 1.5X faster $I_{Na}$ inactivation and 3X max $I_{to}$ conductance and severe Brugada case was modeled with 3.5X faster $I_{Na}$ inactivation and 7X maximum $I_{to}$ conductance. Simulated $BacNa_v$ current was incorporated into the cable at the 0.2X and 0.5X conductance levels as described. The virtual ECG electrode was placed 2 cm away from the epicardium along the fiber axis[35]. The pseudo-ECG signal was calculated using the following integral expression taken from Plonsey and Barr[94]:

$$\phi_e(x', y', z') = \frac{a^2 \sigma_i}{4\sigma_e} \int (-\nabla V_m) \cdot \left[\nabla \frac{1}{r}\right] dx \quad (5)$$

Where $(x',y',z')$ is the location in Euclidean space of the simulated point electrode, $a$ is the radius of the fiber (10 μm), $\sigma_i$ and $\sigma_e$ are the intracellular and extracellular conductivity, respectively, and $r$ is the Euclidean distance from the source point $(x,y,z)$ to the simulated point electrode. For each simulated Brugada case, ECG deviation <ECG − $ECG_{Healthy}$> was calculated by taking the sum of absolute voltage differences overall time points between the Brugada ECG waveform and the healthy ECG waveform.

**AAV production and titration.** All recombinant AAV viruses were generated using the standard triple transfection method as described previously[95]. Specifically, 293T cells (ATCC, CRL-3216) were co-transfected with the adenoviral helper plasmid pALD-X80 (Aldevron), the packaging plasmid AAV2/9 (gift from James M. Wilson, Addgene plasmid #112865), and the transfer ITR plasmid (1:1:1 molar ratios) using polyethylenimine (PEI) 40K Max transfection reagent (Polysciences). Transfected cells were supplied with fresh media 48–72 h after transfection and both cells and supernatant containing virus particles were collected 120 h after transfection. Collected cells were centrifuged (500 × g, 10 min) and the cell pellet was resuspended in cell lysis buffer (0.15 M NaCl + 0.05 M Tris-HCl, pH 8.5) and lysed through four sequential freeze-thaw cycles (15 min in dry ice/ethanol bath followed by 5 min in 37 °C water bath). AAV-containing cell lysate was collected following centrifugation at 3900 × g and 4 °C for 30 min to remove cell debris. Collected media supernatant was filtered through 0.45 mm cellulose acetate filter (Corning) before being combined with 40% polyethylene glycol (PEG) solution at 4:1 volume ratio for overnight incubation at 4 °C. Concentrated AAV particles were harvested following 15 min centrifugation (2818 × g, 4 °C), resuspended in cell lysis buffer, and combined with viral particles collected from the cell pellet. Benzonase (Millipore Sigma) was added to the virus-containing solution at a final concentration of 50 U/ml with subsequent incubation at 37 °C for 30 min. Viral particles were purified via iodixanol density gradient[96] ultracentrifugation at 166,880 × g and 17 °C for 15–17 h (WX Ultra 80, Thermo Fisher Scientific). Fractions containing AAV9 were collected and subjected to subsequent phosphate-buffered saline (PBS) buffer exchange using Zeba Spin (40-kDa-molecular-weight cutoff [MWCO]) desalting columns (Thermo Fisher Scientific). Viral titers of purified viruses were determined by quantitative PCR with primers that specifically amplify the AAV2 ITR regions (forward primer, 5'-AACATGCTACGCAGAGAGGGAGTGG-3'; reverse primer, 5'-CATGAGACA AGGAACCCCTAGTGATGGAG-3') (Integrated DNA Technologies).

**Mouse tail-vein injection.** All mice were housed in 12 h light/dark cycles, at ambient temperatures of 68–79 degrees Fahrenheit, at a humidity range between 30 and 70%, and with access to food and water ad libitum. Male 6–10-week-old CD-1 mice (Charles River Laboratories) were injected via tail vein with 200 μl of AAV9 solution ($2 \times 10^{12}$ vg/mouse for AAV9-CAG-h2SheP-2A-GFP, and $1 \times 10^{12}$ vg/mouse for AAV9-MHCK7-h2SheP-HA-2A-GFP, scAAV9-MHCK7-h2SheP-HA and scAAV9-MHCK7-GFP). Mice were euthanized by isoflurane inhalation 4–6 weeks post-injection and the hearts were harvested for cardiomyocyte isolation or histology.

**Isolation of adult mouse ventricular myocytes.** Adult mouse ventricular cardiomyocytes were isolated and cultured according to a previously published Langendorff procedure[97]. Briefly, the heart was excised and enzymatically digested by perfusion of 40 ml prewarmed enzyme solution (Collagenase II 475 U/ml (Worthington), Blebbistatin 15 μmol/L (Stemcell Technologies)) at a rate of 2 ml/min. The collagenase activity was inhibited with fetal bovine serum (FBS) to a final concentration of 10% and the cell suspensions were passed through a 200 μm filter (BD Biosciences). Calcium concentration was gradually restored using 3 intermediate calcium reintroduction buffers (prepared as previously described[3]) and the cells were allowed to settle by gravity each round for 15 min. The final cell pellet was resuspended in a plating medium and plated onto 21 mm diameter Aclar (Ted Pella) coverslips coated with laminin (5 μg/mL, Thermo Fisher Scientific) and incubated for 4 h at 37 °C before patch-clamp studies.

**Mouse electrocardiograms.** All measurements were conducted and analyzed in a blinded fashion. Mice were anesthetized using a volatile anesthetic system with an induction chamber (R5835, RWD Life Science, Dover, Delaware, United States). Electrocardiographic (ECG) measurements were performed using four subdermal leads: I, aVR, aVL, and aVF. ECG parameters, such as RR, PR, QRS, QT, and corrected QT for heart rate (Bazett's QT correction) were measured and/or calculated at baseline and following adrenergic stimulation and ryanodine receptor sensitization with 200 μg/g caffeine and 1 μg/g isoproterenol IP. Rhythm detection was captured by an iWorx-RA-834 Eight Channel 16-bit Data Acquisition System (iWorx, Dover New Hampshire, United States). Data were viewed using a custom-built ECG Analysis Module software program for LabScribe v4.

**SAN dissection for immunostaining**. Hearts from heparinized mice (200U i.p.) were perfusion-fixed with 4% PFA and immersed in 30% (w/v) sucrose overnight. Ventricles were removed and atria were pinned on a PDMS mold and visualized using a stereomicroscope (DFC7000T; Leica). The SAN region was identified using the superior and inferior vena cava, the right atrial appendage, the crista terminalis, and the interatrial septum as landmarks. The SAN preparation including right and left atria were embedded and frozen in OCT compound (VWR) using a dry ice/isoproterenol bath, cut into 10 μm sections using a cryostat (Leica), and immunostained as described below.

**Immunostaining and imaging**. Cell monolayers were fixed in 4% paraformaldehyde (PFA) for 10 min at room temperature. Hearts were fixed and sectioned as described for the SAN tissue. Fixed monolayers or heart sections were permeabilized and blocked in blocking solution (5% chicken serum + 0.1% Triton-X, 30 min). The following primary antibodies (1 h incubation) were used: anti-sarcomeric α-actinin (Sigma, a7811, 1:200), anti-vimentin (Abcam, ab92547, 1:500), anti-Cx43 (LSBio, LS-B9770, 1:300), anti-HCN4 (Alomone, APC-052, 1:200), anti-cardiac troponin T (Abcam, ab45932, 1:200), and anti-HA tag (Cell Signaling Technology, C29F4, 1:200). Secondary antibodies (1 h incubation) included: Chicken anti-Mouse Alexa Fluor 488 (Thermo Fisher Scientific, A-21200/A-21441, 1:200), Chicken anti-Mouse Alexa Fluor 594 (Thermo Fisher Scientific, A-21201/A-21442, 1:200), Chicken anti-Mouse Alexa Fluor 647 (Thermo Fisher Scientific, A-21463, 1:200), Donkey anti-Rabbit Alexa Fluor Plus 594 (Thermo Fisher Scientific, A-32754, 1:200), Donkey anti-Rabbit Alexa Fluor Plus 647 (Thermo Fisher Scientific, A-32795, 1:200), Alexa Fluor 488-conjugated phalloidin (Thermo Fisher Scientific, A12379, 1:300), Alexa Fluor 647-conjugated phalloidin (Thermo Fisher Scientific, A22287, 1:300), DAPI (Sigma, D9542, 1:300). All immunostaining steps were performed at room temperature. Fluorescence images were acquired using inverted fluorescence (Nikon TE2000) or confocal (Leica SP5, Andor Dragonfly) microscope, and processed with ImageJ software.

**Statistics and reproducibility**. All statistical analyses and data plotting were performed using Prism (GraphPad Software Inc.). D'Agostino–Pearson test was used to confirm data normality. Data are presented as mean ± s.e.m. and represent a minimum of three independent experiments with at least three biological and technical replicates unless otherwise stated. For comparisons of two experimental groups, statistical significance was evaluated with a standard unpaired Student $t$-test (two-tailed; $P < 0.05$) or Chi-square test (two-tailed; $P < 0.05$). For multiple-comparison analysis, statistical significance was determined by one-way or two-way ANOVA, followed by Tukey's post-hoc test to calculate $P$-values. Statistical significance was defined as $P < 0.05$ (95% confidence). For all results, the exact $P$-value, number of biological replicates, and statistical test used are reported in figure legends. All shown images are representative of three independent experiments with at least three biological and technical replicates.

**Reporting summary**. Further information on research design is available in the Nature Research Reporting Summary linked to this article.

## Data availability

All data generated and/or analyzed are available within the manuscript and Supplementary information files. Source data are provided as a Source Data file. Source data are provided with this paper.

## Code availability

The MATLAB code used for optical mapping analysis is specific to a custom-built device in the Bursac group and is not of general utility. All modeling codes and mathematical formulae are referenced and provided in the Methods section. A custom code for BacNa$_v$ channel used in modeling studies is presented as Supplementary Information.

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

## Acknowledgements

We thank J. Woodard, J. Pomeroy, and N. Strash for assistance with NRVM culture, A. Helfer and N. Strash for helping with hiPSC-CM studies, G. Devlin and A. Asokan for assistance with AAV production, K. Heman for performing mouse tail-vein injection, M. ter Weele, C. Curtis, and D. Genuit for assistance with tissue sectioning and immunostaining, and S. Xiong for assistance with adult mouse CM isolation. This work was supported by the National Institutes of Health grants HL134764, HL132389, and HL126524 to N.B., 1U01HL143336-01 to C.H., grants from Duke Translating Duke Health Initiative to N.B. and A.P.L. and American Heart Association Predoctoral Fellowship 829638 to R.M.P.

## Author contributions

H.X.N., C.H., A.L., and N.B. designed the project. H.X.N., T.W., D.N., H.Z., R.P., S.D., R.Y., and M.P. performed experiments. H.X.N., T.W., and N.B. analyzed data and wrote the manuscript.

## Competing interests

H.X.N., T.W., and N.B. are inventors on a pending patent concerning application of prokaryotic sodium channels (PCT:WO2021076600A1). The remaining authors declare no competing interests.
