## [Peer Review File · Nature Communications]

REVIEWER COMMENTS

Reviewer #1 (Remarks to the Author):

This manuscript looked to address a novel gene therapy approach for the therapeutic treatment of compromised cardiac excitability found in both genetic and acquired disease states and its role in arrhythmia induction. The voltage-gated cardiac sodium channel Nav1.5 is responsible for the rapid depolarization and action potential initiation in cardiac excitability and losses in its activity underlies increased susceptibility to lethal arrhythmias found in genetic conditions such as Brugada Syndrome and acquired disease states such as ischemic and hypertrophic remodeling. Currently, there are no therapeutic approaches for restoring losses in Nav1.5 current/excitability and even gene therapy approaches for delivery of exogenous Nav1.5 are not available due to its large size and limited packaging sizes of lentiviral/AAV constructs. This study shows the successful circumvention of this problem by cloning in bacterial sodium channel, which assembles as a homotetramer, allowing for assembly of functional Na⁺ channels and current with the opportunity for packaging into translational viral gene delivery constructs. The authors went through rigorous design of construct promoters for cardiac myocyte specific expression with codon optimization for mammalian expression to obtain high levels of transgene expression. They characterized functional expression and localization of these channels in both in vitro and in vivo settings to show successful restoration of cardiac excitability and reductions in arrhythmic burden. They also utilized computational studies to show thresholds for transgene bacterial Na⁺ channel expression in overcoming arrhythmic substrates like conduction block in settings of increased fibrosis. Overall, this study rigorously designed and tested a creative gene therapy approach for restoring cardiac excitability and showed its ability to reduce arrhythmias, providing novel therapeutic opportunity for conditions where the only course of intervention has been defibrillator implantation. However, there are additional critical studies which would be required to understanding the full potential and implementation of such a therapy.

Major Concerns:

1. The authors focused on co-cultures of neonatal myocytes with fibroblasts to simulate increased fibrosis seen in diseased hearts and compromised cardiac excitability. There were able to successfully show that delivery of bacterial Na⁺ channel increased cellular excitability and reduce the incidence of conduction block, reducing sustained re-entry in these culture dishes. It would be helpful to also see not just structural changes provided by "fibrosis" but also the role of electrical remodeling seen within myocytes with disease. For example, chronic adrenergic signaling has been shown to reduce cellular excitability by reducing expression of native Nav1.5. This effect has been simulated in culture conditions with delivery of agents such as isoproterenol or phenylephrine, which increase arrhythmic burden. Therefore, it would be helpful to see similar therapeutic treatment within culture conditions following prolonged exposure to adrenergic agents.

2. Significant focus was given to in vitro characterization of the effects of bacterial Na⁺ channel expression. While in vivo delivery was performed, it was only used to characterize expression and localization of the bacterial channel in adult cardiomyocytes. It would be helpful to see the impact of in vivo delivery of bacterial Na⁺ channel on baseline electrical parameters through surface ECG analysis of these animals. What impact does delivery of this construct have on heart rate (R-R interval) or QRS duration with increased excitability? Does this construct express within the sinoatrial node or other pacemaker/conduction cell types which could disrupt electrical behavior of the intact heart.

3. Similar to the previous aim which mentions just baseline characteristics of electrical properties, it would be beneficial to see the electrical impact (ECG recordings) of isoproterenol and/or co-delivery with caffeine in animals with the bacterial Na⁺ channel. The combination of the drugs have been shown to be pro-arrhythmic through increasing premature ventricular contractions (PVCs) through disruptions in Ca²⁺ handling. Would increasing cellular excitability actually increase the occurrence of such PVCs but reduce sustained arrhythmias? Given that acquired disease in the heart expresses both reduced excitability but also disruptions in Ca²⁺ handling it is critical to know the impact of restoring or even achieving super-physiologic levels of excitability and its impact on sustained arrhythmias.

4. This gene therapy approach also represents a potential treatment for genetic conditions of reduced cardiac excitability, such as Brugada Syndrome. It would therefore be helpful to observe in vivo characterization of delivery of bacterial Na⁺ channel in established mouse models of Brugada Syndrome to show its therapeutic potential.

5. When confirming gene delivery to mouse ventricular myocytes from tail vein injections, they only examined expression of their transfected construct one week after tail vein injection. I would have liked to see at least one longer time point, as that would give a clearer picture of how long the transfected BacNav hangs around. They do show representative current traces from IBacNav, but they show those at the 4 week time point and the immunostaining at the 1-week timepoint. Their case that in vivo gene delivery of BacNav yields expression of functional channels (and that those hang around for a while) would just be stronger if they show both the immunostaining images and the I-V curves for both 1 week and 4 week time points.

Reviewer #2 (Remarks to the Author):

NCOMMS-20-22385 Nguyen et al.

Nguyen et al. report further optimization and implementation of using bacterial sodium channels (BacNaVs) as a means to modify the electrical properties of targeted cells, specifically cardiomyocytes. This manuscript presents an optimized BacNaV (NaVSheP D60A) having improved expression properties and shows that this channel can reduce conduction block and reentrant arrhythmias in fibrotic cardiac cultures. The authors also show that they can achieve expression of this BacNaV in the mouse hearts following injection of an adenovirus vector bearing the channel. This last step is a key advance for implementing this class of channels as a potential gene therapy.

Overall, the study is well done. There are a number of presentation gaps that need to be addressed.

Intro line 67 . There are hundreds of BacNaVs. It is unclear from this description which or how many were used. Authors may have intended to say 'we demonstrated that a BacNaV...'

Same issue on line 78. The definite (or indefinite) article is missing before 'BacNaV'.

Results: lines 84-85 NaVSheP should be referenced (original Irie paper, JBC 285:3685-3594 (2010)). The channel used here is aa mutant, described in Ref. 12, and should be both clearly described and referenced. As currently written, it is not clear if the D60A represents a mutant, or whether this is part of the name of the channel. Refernces for for SheP should also be included in the Methods section (line 325)

Exemplar currents, not just the processed data should be shown in Fig. 1.

It is not clear what 'Expression' means on the y-axis of Fig. 3 a. What is measured? What are the units? One has to wade through the text to see that this is normalized expression to GAPDH. That fact should be on the axis, but it is not apparent from the text or figure what has been measured and compared. Figure 4, Supplementary Fig. 4 have the same issue.

The section on trafficking is a bit confusing. Basically, the authors find that introduction of the GFP tag causes trafficking problems to hSheP, that they then overcome by using an ER export motif, but then find that the channel alone having a simple HA tag does not have this trafficking problem. So, they make a problem, fix it, and then find out that fixing it is not necessary. This provides a rationale for avoiding the GFP tag in later applications, but makes for a very strange diversion. I can see why these facts may be important, but the presentation could be clearer.

Reviewer #3 (Remarks to the Author):

The manuscript describes the expression of prokaryotic BacNav in cardiac myocytes as a potential to increase cardiac excitability and conduction as a therapy for cardiac arrhythmias. The relatively small size of BacNav can be comfortably accommodated within AAV vectors, in contrast to the large size of mammalian Nav1.5 which exceeds the packaging capacity of viral vectors used for gene therapy. The authors use codon-optimization and viral vectors to robustly express BacNav in various cardiac myocytes preparations as well as engineered excitable HEK293 cells. In vitro experiments carried out on monolayer cultures of neonatal rat ventricular myocytes, as well as in silico simulations, suggested BacNav expression could be beneficial in reducing some cardiac arrhythmias by enhancing excitability and reducing reentry.

Overall, the idea of expressing prokaryotic channels in mammalian cells as a tool to control excitability in mammalian cells is an interesting concept and one that is likely to have some applications. However, in its current form the potential of using BacNav in gene therapy for cardiac arrhythmias seems rather far-fetched due to limitations (slow gating kinetics) that the authors fail to address in the manuscript.

1. Both the amplitude and gating kinetics (fast activation and inactivation) of Nav1.5 channels are critical for proper excitation and performance of the heart. The relatively slow activation and inactivation kinetics of BacNav would appear to be problematic for potential therapeutic applications in the heart. Mutations in Nav1.5 that slow inactivation yield late currents, prolong the APD, result in long QT syndrome, and are pro-arrhythmic. The slow inactivation of BacNav would essentially act as late current and likely to produce a pro-arrhythmic effect in vivo. This likely adverse effect of BacNav due to its slow gating kinetics is not considered in the manuscript.

2. It is somewhat surprising that the simulations of the impact of BacNav expression on the human/guinea pig/dog cardiomyocytes did not show an increase in the APD given the introduction

of an extra inward positive current. This is not entirely consistent with observations that Nav1.5 mutations that produce late current result in long QT syndrome. This particular prediction should be directly tested experimentally as it directly relates to whether gene therapy with BacNav in its current form is a viable potential therapy for cardiac arrhythmias.

3. The in vivo gene expression of BacNav in heart by tail-vein inject is a nice feature of the study. The impact of BacNav on the excitability of adult cardiomyocytes is critical, but this important aspect of the work was not pursued in Fig. 7. A more thorough electrophysiological analyses of the impact of BacNav expression on mouse adult cardiomyocytes (e.g action potentials) should be done. This would also potentially address point 2 above.

We very much appreciate insightful comments of the reviewers that we believe have resulted in a significantly improved manuscript. We also apologize to reviewers for a long time it took us to revise the manuscript. This has been the result of extensive turnaround of personnel during the COVID pandemic. Reviewers' comments are cited in bold followed by our answers shown in normal font. Related changes in the main text of the revised manuscript are highlighted in red. Other minor changes in the text are made to improve readability and satisfy formatting requirements of *Nature Communications*.

REVIEWER COMMENTS

Reviewer #1

Major Concerns:

1. The authors focused on co-cultures of neonatal myocytes with fibroblasts to simulate increased fibrosis seen in diseased hearts and compromised cardiac excitability. There were able to successfully show that delivery of bacterial Na⁺ channel increased cellular excitability and reduce the incidence of conduction block, reducing sustained re-entry in these culture dishes. It would be helpful to also see not just structural changes provided by “fibrosis” but also the role of electrical remodeling seen within myocytes with disease. For example, chronic adrenergic signaling has been shown to reduce cellular excitability by reducing expression of native Nav1.5. This effect has been simulated in culture conditions with delivery of agents such as isoproterenol or phenylephrine, which increase arrhythmic burden. Therefore, it would be helpful to see similar therapeutic treatment within culture conditions following prolonged exposure to adrenergic agents.

We appreciate the suggestion by the reviewer. There is one previous *in vitro* study by Pijnappels and colleagues¹ who elegantly explored mechanisms of arrhythmias in: 1) a cardiac fibrosis model (similar to the one used in our study) and 2) a model of phenylephrine-induced abnormalities in calcium handling. We have thus performed similar experiments to examine BacNa_v effects in NRVM cultures exposed to phenylephrine. Results of these studies are shown in new Supplementary Fig. 10 and Video 4. While in agreement with Pijnappels et al. we found that phenylephrine treatment increased CM size, APD, and incidence of triggered focal arrhythmias, no conduction slowing was found. Furthermore, BacNa_v expression in phenylephrine-treated cultures showed a trend toward reduced arrhythmia incidence, but statistical significance was not reached. Importantly, the incidence of arrhythmias was not increased. This confirms that the primary antiarrhythmic utility of BacNa_v would be in the settings of conduction slowing and fibrosis rather than calcium handling abnormalities. The new results in response to this comment, related methods, and discussion are now described in the revised manuscript in the following lines: 259-270; 370-376; 458-460

2. Significant focus was given to in vitro characterization of the effects of bacterial Na⁺ channel expression. While in vivo delivery was performed, it was only used to characterize expression and localization of the bacterial channel in adult cardiomyocytes. It would be helpful to see the impact of in vivo delivery of bacterial Na⁺ channel on baseline electrical parameters through surface ECG analysis of these animals. What impact does delivery of this construct have on heart rate (R-R interval) or QRS duration with increased excitability? Does this construct express within the sinoatrial node or other pacemaker/conduction cell types which could disrupt electrical behavior of the intact heart.

To address this comment, we have performed new experiments to evaluate surface ECGs in mice injected with either scAAV9-MHCK7-GFP or scAAV9-MHCK7-h2SheP-HA virus. Results of these studies are shown in new Supplementary Fig. 11a-b. Six weeks following AAV delivery, we found no effects of h2SheP on ECG morphology, heart rate, or other measured ECG parameters. We then quantified expression of transgenes (from immunostaining for HA and GFP) in ventricles, atria, and

sinoatrial node (SAN). Consistent with no change in heart rate induced by BacNav, we found minimal transgene expression in the SAN compared to ventricles and atria. These results are shown in new Fig. 7. To our knowledge, this is the first time that distribution of AAV9-MHCK7-driven gene expression has been systematically evaluated in the heart, and the SAN in particular. We believe that this information is highly relevant for emerging human gene therapies for striated muscle diseases that utilize this combination of the AAV9 capsid and MHCK7 promoter (e.g. clinical trials for Duchenne Muscular Dystrophy). We thus very much appreciate this comment of the reviewer. The new results in response to this comment, related methods, and discussion are now described in the revised manuscript in lines: 276-281; 286-294; 344-353; 616-631

3. Similar to the previous aim which mentions just baseline characteristics of electrical properties, it would be beneficial to see the electrical impact (ECG recordings) of isoproterenol and/or co-delivery with caffeine in animals with the bacterial Na⁺ channel. The combination of the drugs have been shown to be pro-arrhythmic through increasing premature ventricular contractions (PVCs) through disruptions in Ca²⁺ handling. Would increasing cellular excitability actually increase the occurrence of such PVCs but reduce sustained arrhythmias? Given that acquired disease in the heart expresses both reduced excitability but also disruptions in Ca²⁺ handling it is critical to know the impact of restoring or even achieving super-physiologic levels of excitability and its impact on sustained arrhythmias.

We thank the reviewer for the suggestion. We performed co-delivery of caffeine and isoproterenol to mice injected with either scAAV9-MHCK7-GFP or scAAV9-MHCK7-h2SheP-HA virus and measured changes in ECGs (results shown in new Supplementary Fig. 11c). We did not see any significant difference in any of the measured ECG parameters. We also did not observe any PVCs or arrhythmias in any of tested animals. In our experience, induction of PVCs in normal mouse hearts even with drug treatment will be at the best case sporadic, and inducing sustained ventricular arrhythmias will be very difficult due to small mouse heart size. Importantly, as in our *in vitro* studies, we do not see that BacNav expression results in the induction of pro-arrhythmic effects. The new results in response to this comment, related methods, and discussion are now described in the revised manuscript in lines: 281-286; 616-624

4. This gene therapy approach also represents a potential treatment for genetic conditions of reduced cardiac excitability, such as Brugada Syndrome. It would therefore be helpful to observe in vivo characterization of delivery of bacterial Na⁺ channel in established mouse models of Brugada Syndrome to show its therapeutic potential.

We agree with the reviewer that studying BacNav effects in yet another model of arrhythmia, such as Brugada Syndrome, would also be of interest. Currently, there exist two well-characterized transgenic mouse models that show generic features of Brugada Syndrome (to our knowledge, both of them are only available in Europe). The first model is heterozygous knockout of Nav1.5 (SCN5A^{+/-}) which results in a ~50% loss in sodium current²; and the second model is Scn5a-1798insD knock-in model which shows overlap of Long QT syndrome 3 (LQT3), Brugada syndrome, and progressive cardiac conduction defect³. Both of the models, while valuable for the research community, are variable in their phenotype and do not exhibit human Brugada type pattern in the surface ECG, mainly because of the short mouse cardiac action potential and high heart rate compared to those of humans or large animals.

In our original manuscript, we have shown *in vitro* (Fig. 4f-h) and *in silico* (Fig. 5a-f, Supplementary Fig. 7a,b) beneficial effects of BacNav on CMs (human, rat, dog) with reduced sodium current, akin to Brugada phenotype. To address this reviewer's comment, we have additionally adapted a Brugada syndrome model (T1620M mutation in SCN5A)⁴ and tested the effect of BacNav expression on simulated AP shape and pseudo-ECG in a model of cardiac transmural conduction. Results of these studies are shown in new Fig. 5j-o. Specifically, we modeled mild and severe Brugada syndrome cases via specific changes in I_{Na} inactivation and I_{to} amplitude. In both cases, the expression of BacNav rescued the decreased phase 0 amplitude and increased phase 1 dip in midmyocardium and

epicardium (Fig. 5l,m), the loss of AP dome in epicardium (Fig. 5m), and the characteristic ST-segment elevation in pseudo-ECG (Fig. 5n,o). Overall, due to limited availability and, to some extent, relevance of the transgenic mouse models of Brugada syndrome, we opted to show potential beneficial effects of BacNav_v expression in multiple Brugada-relevant settings using *in vitro* and *in silico* models. While the results of these studies are promising, demonstrating clinically relevant effects of BacNav_v therapy in Brugada syndrome would be more meaningful in pharmacologically-induced or genetically engineered large animal models of disease, which is outside the scope of our current manuscript. The new results in response to this comment, related methods, and discussion are now described in the revised manuscript in lines: 226-239; 554-574

5. When confirming gene delivery to mouse ventricular myocytes from tail vein injections, they only examined expression of their transfected construct one week after tail vein injection. I would have liked to see at least one longer time point, as that would give a clearer picture of how long the transfected BacNav hangs around. They do show representative current traces from IBacNav, but they show those at the 4 week time point and the immunostaining at the 1-week timepoint. Their case that in vivo gene delivery of BacNav yields expression of functional channels (and that those hang around for a while) would just be stronger if they show both the immunostaining images and the I-V curves for both 1 week and 4 week time points.

Thank you for this comment. Immunostainings and corresponding quantification in new Fig. 7a-d now demonstrate channel expression in hearts 6 weeks after virus injection. To further address the reviewer's comment, we have additionally included both immunostaining images and I-V curves from the same batches of CMs isolated 4 weeks after virus injection (shown in new Fig. 8a-f). The new results in response to this comment, related methods, and discussion are now described in the revised manuscript in lines:274-276; 295-302

Reviewer #2:

1. Intro line 67 . There are hundreds of BacNaVs. It is unclear from this description which or how many were used. Authors may have intended to say 'we demonstrated that a BacNaV....'. Same issue on line 78. The definite (or indefinite) article is missing before 'BacNaV'.

We appreciate this comment and now specify BacNa_v variant names in the revised manuscript in lines: 76-77; 88-89

2. Results: lines 84-85 NaVSheP should be referenced (original Irie paper, JBC 285:3685-3594 (2010)). The channel used here is aa mutant, described in Ref. 12, and should be both clearly described and referenced. As currently written, it is not clear if the D60A represents a mutant, or whether this is part of the name of the channel. Refernces for for SheP should also be included in the Methods section (line 325)

Na_vSheP D60A is indeed a mutated version of the wild-type Na_vSheP channel first described by Irie et al. in their 2010 paper. We have now clearly described and referenced used BacNa_v channels in the revised manuscript in lines: 95-96; 407

3. Exemplar currents, not just the processed data should be shown in Fig. 1.

Thank you for this comment. We now include in revised Figure 1 representative current (Fig. 1e) and action potential (Fig. 1g) traces for all three groups.

4. It is not clear what 'Expression' means on the y-axis of Fig. 3 a. What is measured? What are the units? One has to wade through the text to see that this is normalized expression to GAPDH. That fact should be on the axis, but it is not apparent form the text or figure what has been measured and compared. Figure 4, Supplementary Fig. 4 have the same issue.

We appreciate this comment. The y-axes of all figures showing qPCR data are now edited to specify the housekeeping gene that the expression is normalized to. These include Figures 2d, 3a, 4a, and Supplementary Figure 4.

5. The section on trafficking is a bit confusing. Basically, the authors find that introduction of the GFP tag causes trafficking problems to hSheP, that they then overcome by using an ER export motif, but then find that the channel alone having a simple HA tag does not have this trafficking problem. So, they make a problem, fix it, and then find out that fixing it is not necessary. This provides a rationale for avoiding the GFP tag in later applications, but makes for a very strange diversion. I can see why these facts may be important, but the presentation could be clearer.

We appreciate the reviewer's comment. Our results demonstrate that adding an ER export motif can improve trafficking of GFP-tagged BacNa_v, but we agree with the reviewer that the result can be confusing while not contributing essentially to other findings. We thus decided to remove this figure from the revised manuscript.

Reviewer #3:

1. Both the amplitude and gating kinetics (fast activation and inactivation) of NaV1.5 channels are critical for proper excitation and performance of the heart. The relatively slow activation and inactivation kinetics of BacNav would appear to be problematic for potential therapeutic applications in the heart. Mutations in Nav1.5 that slow inactivation yield late currents, prolong the APD, result in long QT syndrome, and are pro-arrhythmic. The slow inactivation of BacNav would essentially act as late current and likely to produce a pro-arrhythmic effect in vivo. This likely adverse effect of BacNav due to its slow gating kinetics is not considered in the manuscript.

We very much appreciate this comment. Despite slower gating kinetics than Na_v1.5, we have shown that BacNa_v (h2SheP) expression does not prolong APD in CMs when endogenous APD lasts 100 ms or longer. This is evident from our studies in NRVMs (Fig. 2f, Fig. 4i, and new Supplementary Fig. 10d), human iPSC-CMs (Supplementary Fig. 5f), simulated guinea pig ventricular CMs (Fig. 5k-m, Supplementary Fig. 7b), simulated dog ventricular CMs (Supplementary Fig. 7a), and simulated human ventricular CMs (Fig. 5b,d). Furthermore, in none of the *in vitro* or *in silico* cardiac cell or tissue experiments have we observed that BacNa_v expression led to any arrhythmogenic events, including EADs, previously reported for the late Na_v1.5 current^{5, 6}.

To further address the reviewer's comment, we have performed AP-clamp recordings in our 2 excitable HEK293 lines: 1) the Ex293 line expressing Na_v1.5+Kir2.1+Cx43 and 2) the KirCxSheP293 line expressing h2SheP+Kir2.1+Cx43. In the Ex293 line, we also applied 100 μM ATX_II to induce late Na_v1.5 current^{7, 8}. We then applied simulated human AP as the command potential in the voltage-clamp mode and observed that compared to KirCxSheP293 cells where the BacNa_v current fully turned off in <50 ms, in Ex293 cells the ATX_II induced late Na_v1.5 current persisted and increased in amplitude during late repolarization (new Supplementary Fig. 3a-e), consistent with previous studies⁸. In the current-clamp mode, we observed abnormally long APs in the presence of ATX_II, as previously reported⁸⁻¹⁰, but not in the Ex293 or KirCxSheP293 cells (new Supplementary Fig. 3f-h). Based on these studies, we believe that BacNa_v current, which turns off relatively early during AP, will have qualitatively different effects from late Na_v1.5 current, which persists into late phases of AP. These results, along with no APD prolongation observed in our studies for APs lasting >100ms (please see also our response to comment #2 below) and no increase in triggered activity in phenylephrine-treated NRVMs (new Supplementary Fig. 10f), suggest that BacNa_v expression in the human heart would not yield pro-arrhythmic effects characteristic of late I_{Na}. These results and related discussion are now described in the revised manuscript in lines: 157-169; 259-270; 377-388

It is also worth noting that BacNa_vs are simple homotetrameric channels that lack elaborate intracellular loops and thus may be less prone to post-translational modifications compared to Na_v1.5 channels, such as phosphorylation by CaMKII on serine 571 in the IDI-II linker which may regulate the late I_{Na} component of Na_v1.5¹¹. Hence, it is plausible that the simpler structure of BacNa_v not only makes stable gene delivery feasible, but may also protect BacNa_v channels from abnormal regulation under pathological conditions. We plan to explore these questions in our future studies.

2. It is somewhat surprising that the simulations of the impact of BacNav expression on the human/guinea pig/dog cardiomyocytes did not show an increase in the APD given the introduction of an extra inward positive current. This is not entirely consistent with observations that Nav1.5 mutations that produce late current result in long QT syndrome. This particular prediction should be directly tested experimentally as it directly relates to whether gene therapy with BacNav in its current form is a viable potential therapy for cardiac arrhythmias.

We thank the reviewer for the insightful comment. As shown in our experiments (Fig. 4f-i) and simulations (Fig. 5 and Supplementary Fig. 7), the additional inward current through BacNa_v channels yields increased action potential amplitude and early plateau potential. These higher potentials (V_m) result in more delayed rectifier K⁺ channels opening and a larger driving force (V_m-E_k) to generate more

outward/repolarizing current, which opposes the extra inward BaNa_v current, thus preventing APD prolongation. To support this idea, we added a new Supplementary Fig. 7c showing rapid and slow delayed rectifier K^+ currents (I_{Kr} and I_{Ks}) during simulated human, dog, and guinea pig APs under different BaNa_v expression levels. The BaNa_v expression yielded both earlier activation and increased magnitude of I_{Ks} in all models. These results and related discussion are now described in the revised manuscript in lines: 199-203; 377-388

Furthermore, as mentioned in the response to comment #1, our optical mapping and patch clamp recordings also demonstrated that BaNa_v expression did not result in increase in APD in NRVMs (Fig. 2f, Fig. 4i, and new Supplementary Fig. 10d), human iPSC-CMs (Supplementary Fig. 5f), or simulated ventricular CMs (Fig. 5b,d,k-m, Supplementary Fig. 7a,b). On the other hand, in cells with short-lasting action potentials (<100ms) such as Ex293 cells (Supplementary Fig. 2b,c) or serum-free cultured NRVMs (Fig. 6e), BaNa_v expression prolonged the APD, potentially because these cells may not have robust delayed rectifier K^+ currents. Please see also our response to comment #3 below.

3. The in vivo gene expression of BaNav in heart by tail-vein inject is a nice feature of the study. The impact of BaNav on the excitability of adult cardiomyocytes is critical, but this important aspect of the work was not pursued in Fig. 7. A more thorough electrophysiological analyses of the impact of BaNav expression on mouse adult cardiomyocytes (e.g action potentials) should be done. This would also potentially address point 2 above.

We appreciate this important comment. We have now performed current-clamp AP recordings in healthy isolated ventricular mouse CMs 6 weeks following AAV9- BaNa_v -GFP injection (results shown in new Fig. 8g-m). While we observed a trend towards a higher maximum AP upstroke, the only significant change due to BaNa_v expression was the increased APD_{90} , consistent with what we observed *in vitro* in other cell types with short APD (discussed above). No change in AP amplitude despite BaNa_v expression was likely a result of large I_{to} present in mouse CMs that opposed BaNa_v current. This in turn may have prevented BaNa_v -induced increase in repolarizing K^+ currents explained above (Supplementary Fig. 7c), yielding longer APD_{90} . On the other hand, no QT prolongation observed in surface ECGs (new Supplementary Fig. 10b), despite patch-clamp measured increase in APD_{90} , is likely a consequence of high resting mouse heart rate, at which BaNa_v channels are largely inactive (as shown in our previous publication, Nguyen et al. 2016, Nature Comm, Supplementary Fig. 7g-i). Overall, we believe that due to their fast heart rate and short AP, mice are not suitable model for evaluating therapeutic efficacy of BaNa_v *in vivo*. We intend to perform further experiments in larger animals where heart rate and AP shape better align with human physiology. The new results in response to this comment, related methods, and discussion are now described in the revised manuscript in lines: 302-305; 377-399

Reference

1. Askar, S.F. et al. Similar arrhythmicity in hypertrophic and fibrotic cardiac cultures caused by distinct substrate-specific mechanisms. *Cardiovasc Res* **97**, 171-181 (2013).
2. Papadatos, G.A. et al. Slowed conduction and ventricular tachycardia after targeted disruption of the cardiac sodium channel gene *Scn5a*. *Proc Natl Acad Sci U S A* **99**, 6210-6215 (2002).
3. Remme, C.A. et al. Overlap syndrome of cardiac sodium channel disease in mice carrying the equivalent mutation of human *SCN5A-1795insD*. *Circulation* **114**, 2584-2594 (2006).
4. Dumaine, R. et al. Ionic mechanisms responsible for the electrocardiographic phenotype of the Brugada syndrome are temperature dependent. *Circ Res* **85**, 803-809 (1999).
5. Shryock, J.C., Song, Y., Rajamani, S., Antzelevitch, C. & Belardinelli, L. The arrhythmogenic consequences of increasing late I_{Na} in the cardiomyocyte. *Cardiovasc Res* **99**, 600-611 (2013).
6. Maltsev, V.A., Silverman, N., Sabbah, H.N. & Undrovinas, A.I. Chronic heart failure slows late sodium current in human and canine ventricular myocytes: implications for repolarization variability. *Eur J Heart Fail* **9**, 219-227 (2007).

7. Jia, S. et al. Modulation of the late sodium current by ATX-II and ranolazine affects the reverse use-dependence and proarrhythmic liability of IKr blockade. *Br J Pharmacol* **164**, 308-316 (2011).
8. Horvath, B. et al. Dynamics of the late Na(+) current during cardiac action potential and its contribution to afterdepolarizations. *J Mol Cell Cardiol* **64**, 59-68 (2013).
9. Song, Y., Shryock, J.C., Wu, L. & Belardinelli, L. Antagonism by ranolazine of the pro-arrhythmic effects of increasing late INa in guinea pig ventricular myocytes. *J Cardiovasc Pharmacol* **44**, 192-199 (2004).
10. Boutjdir, M. & el-Sherif, N. Pharmacological evaluation of early afterdepolarisations induced by sea anemone toxin (ATXII) in dog heart. *Cardiovasc Res* **25**, 815-819 (1991).
11. Glynn, P. et al. Voltage-Gated Sodium Channel Phosphorylation at Ser571 Regulates Late Current, Arrhythmia, and Cardiac Function In Vivo. *Circulation* **132**, 567-577 (2015).

REVIEWERS' COMMENTS

Reviewer #1 (Remarks to the Author):

The authors have done an excellent job in addressing concerns in the revised manuscript.

Reviewer #2 (Remarks to the Author):

The authors have addressed my concerns. I have no further suggestions for the manuscript.

Reviewer #3 (Remarks to the Author):

The authors have responded comprehensively to the previous critiques and are to be commended for the work they have done to address the concerns raised. The manuscript is much improved and makes a nice contribution.